# Bidirectional control of fear memories by cerebellar neurons projecting to the ventrolateral periaqueductal grey

Jimena Laura Frontera[1], Hind Baba Aissa[1], Romain William Sala[1], Caroline Mailhes-Hamon[1], Ioana Antoaneta Georgescu [1], Clément Léna[1] & Daniela Popa [1✉]

Fear conditioning is a form of associative learning that is known to involve different brain areas, notably the amygdala, the prefrontal cortex and the periaqueductal grey (PAG). Here, we describe the functional role of pathways that link the cerebellum with the fear network. We found that the cerebellar fastigial nucleus (FN) sends glutamatergic projections to vlPAG that synapse onto glutamatergic and GABAergic vlPAG neurons. Chemogenetic and optogenetic manipulations revealed that the FN-vlPAG pathway controls bi-directionally the strength of the fear memories, indicating an important role in the association of the conditioned and unconditioned stimuli, a function consistent with vlPAG encoding of fear prediction error. Moreover, FN-vlPAG projections also modulate extinction learning. We also found a FN-parafascicular thalamus pathway, which may relay cerebellar influence to the amygdala and modulates anxiety behaviors. Overall, our results reveal multiple contributions of the cerebellum to the emotional system.

[1] Neurophysiology of Brain Circuits Team, Institut de Biologie de l'Ecole normale supérieure (IBENS), Ecole normale supérieure, CNRS, INSERM, PSL Research University, 75005 Paris, France. ✉email: daniela.popa@bio.ens.psl.eu

Fear conditioning is a form of associative learning in which an animal learns to associate the presence of a neutral stimulus (conditioned stimulus, CS), with the presence of an aversive fear eliciting stimulus (unconditioned stimulus, US). The amygdala complex, comprised of the basolateral complex (BLA), the central nucleus (CEA), and the intercalated cell masses, is one of the key brain regions for the fear memory acquisition and storage[1–4]. Fear conditioning induces changes not only into the amygdala, but also in auditory and multimodal nuclei of the thalamus, auditory cortex, medial prefrontal cortex (mPFC), and hippocampus[2,5–8]. In addition, fear extinction involves the formation of a new memory trace that attenuates fear responses to a conditioned aversive memory. The distributed network that controls fear extinction involves many of the brain areas that are important for fear conditioning[2,7,9].

The midbrain periaqueductal gray (PAG) is known to mediate both learned and innate freezing behavior via excitatory projections to the magnocellular medulla from the ventrolateral PAG (vlPAG)[10]. In learned freezing, the recruitment of these neurons is produced by a disinhibition triggered by GABAergic inputs from the amygdala in cued fear[10], and glutamatergic inputs from the mPFC in contextual fear[6]. Some vlPAG projection neurons also participates to pain processing[11–13]. However, vlPAG is not a simple regulator of sensory and motor functions, it is also involved in fear learning by generating a "fear prediction error" assessing how unexpected is an incoming US[14]; this step is essential in implementing the Rescorla–Wagner rule which states that learning is driven by departure from expectation[15].

Despite the well-described role of the cerebellum in motor coordination and sensorimotor integration, it is now well established that the cerebellum is also involved in a number of non-motor functions, e.g., refs. [16–19]. Notably, the cerebellum is strongly recruited in aversive emotional states: in humans, neuroimaging studies have revealed changes in cerebellar activation, also mainly in the vermis, in relation to negative emotions (e.g., during recall of self-generated emotional episodes[20]). Consistent with this, localized cerebellar lesions, principally in the midline vermis, account in large part for the emotional disturbances, inappropriate behavior and changes in affect, which are collectively termed the "cerebellar cognitive affective syndrome"[21] and reported in cerebellar patients. Several studies have shown that the cerebellum has functional connections with fear-related areas, including the PAG, the amygdala, and the PFC[22–26]. In accordance with the existence of such connections, Pavlovian fear conditioning affects cerebellar plasticity[27], post-conditioning cerebellar inactivation affects memory consolidation[28], and cerebellar lesions—or inactivation—modulate freezing[24,29].

The cerebellar vermis, which is most consistently associated with emotional pathologies and fear expression[20,24,30], but see ref. [31], projects to the fastigial nucleus (FN), one of the deep cerebellar nuclei that projects to many targets from the spinal cord to the diencephalon[22,32]. The purpose of the present work is to study the contribution of specific FN output pathways to fear learning. Using neuroanatomical tracings, chemogenetic modulation of the cerebellar input to the vlPAG during fear conditioning and extinction, optogenetics and extracellular electrophysiological recordings in awake freely moving animals, we demonstrate the contribution of the cerebellum to fear learning through its inputs to the vlPAG.

## Results

### Neuroanatomical link between the cerebellum and vlPAG.
In order to examine cerebellar projections to areas involved in the fear circuitry, we used anterograde viral tracing and we

have identified fear-related target areas of the projections from the FN. We stereotaxically infused AAV-mCherry in the FN and we found projections exhibiting varicosities into the vlPAG (Fig. 1a–f), which receives inputs from the CEA driving the freezing behavior[10]. Retrograde tracing from vlPAG also indicated the existence of monosynaptic projections from the FN, as well as from the CEA, from the PL and IL areas of the mPFC (Supplementary Fig. 1a–e). Quantification of retrograde labeling from the vlPAG showed that at least $11.2 \pm 0.8\%$ of neurons in the FN projects to vlPAG ($n = 4$; Fig. 1g–i), suggesting that this pathway plays an important contribution to cerebellar control of the emotional system[22]. Moreover, combining a *vglut2-cre* mice line with the infusion of AAV-DIO-tdTom in FN, and retrograde AAV-GFP in vlPAG allowed us to determine that the FN–vlPAG-projecting neurons correspond to vesicular glutamate transporter 2 expressing (vGluT2+) neurons ($85.4 \pm 4.0\%$ of GFP-expressing cells co-localized with vGluT2+ neurons, $n = 4$; Fig. 1j–l). On the other hand, the retrograde tracing in *glyt2-gfp* mice failed to reveal the FN–vlPAG glycinergic-projecting neurons (none of retrograde stained cell co-localized with GlyT2+ neurons in the FN, $n = 3$; Supplementary Fig. 1f, g). Moreover, FN GABAergic neurons are known to only project to the inferior olive[33]. Overall, these results indicate that the vlPAG receives glutamatergic monosynaptic inputs from the FN of the cerebellum.

### FN input onto glutamatergic and GABAergic vlPAG neurons.
To determine whether the inputs from the FN synapse either onto excitatory or inhibitory neurons in the vlPAG, we analyzed the axonal projections and synaptic boutons of FN vGluT2+ neurons. We localized glutamatergic FN terminals by cre-dependent viral expression of the presynaptic marker synaptophysin-GFP in adult *vglut2-cre* mice (Fig. 2a, e). In addition, we identified glutamatergic neurons in the vlPAG by infusion of AAV-DIO-tdTomato, or the GABAergic neurons by immunostaining of GAD67. Glutamatergic FN terminals were found to contact both vGlut2+ and GAD67+ neurons in vlPAG (Fig. 2a–d, e–h, respectively). In fields of vlPAG with dense cerebellar inputs, we found synaptophysin+ boutons on $70.1 \pm 3.0\%$ of GAD67+ cells ($n = 9$ slices from three mice, counted on fields of $8.1\mathrm{e}4\ \mu\mathrm{m}^2$) and on $90.1 \pm 1.1\%$ of vGluT2+ cells ($n = 10$ slices from three mice, Fig. 2i, same field size), from a similar GAD67+ and vGluT2+ cell density area in vlPAG ($419 \pm 54$ GAD67+Syn+ cells mm$^{-2}$, from $613 \pm 86$ GAD67+ cells mm$^{-2}$; and $604 \pm 57$ vGluT2+ Syn+ cells mm$^{-2}$, from $675 \pm 64$ vGluT2+ cells mm$^{-2}$; Fig. 2j). Therefore, these results indicate that FN glutamatergic projections target high proportion of both glutamatergic and GABAergic vlPAG neurons in regions of the vlPAG.

### Fastigial stimulation induces short-latency response in vlPAG.
To study the impact of optogenetic FN stimulation in the vlPAG, we expressed channel-rhodopsin-2 (ChR2) specifically in FN–vlPAG-projecting neurons by local injection of cre-dependent AAV-DIO-ChR2-GFP in the FN, combined with local injection of retrograde AAV-cre-EBFP into the vlPAG (Supplementary Fig. 2a, b). Then, we stimulated FN–vlPAG neurons via FN illumination and we recorded vlPAG activity in awake freely moving animals (Fig. 3a–f). Under FN stimulation, we found LFP negative deflection (Fig. 3a) and cells exhibiting an increase of firing in vlPAG contra-lateral to the FN stimulation (Fig. 3b, c), where the FN is preferentially sending projections. The ramp of stimulation produced a range of vlPAG responses, which increased as a function of light intensity (Fig. 3d). Responses at low intensities were found in the vlPAG contra-lateral to the stimulated FN, while increasing intensities recruited cells on the ipsi-lateral vlPAG (Fig. 3e). The latencies

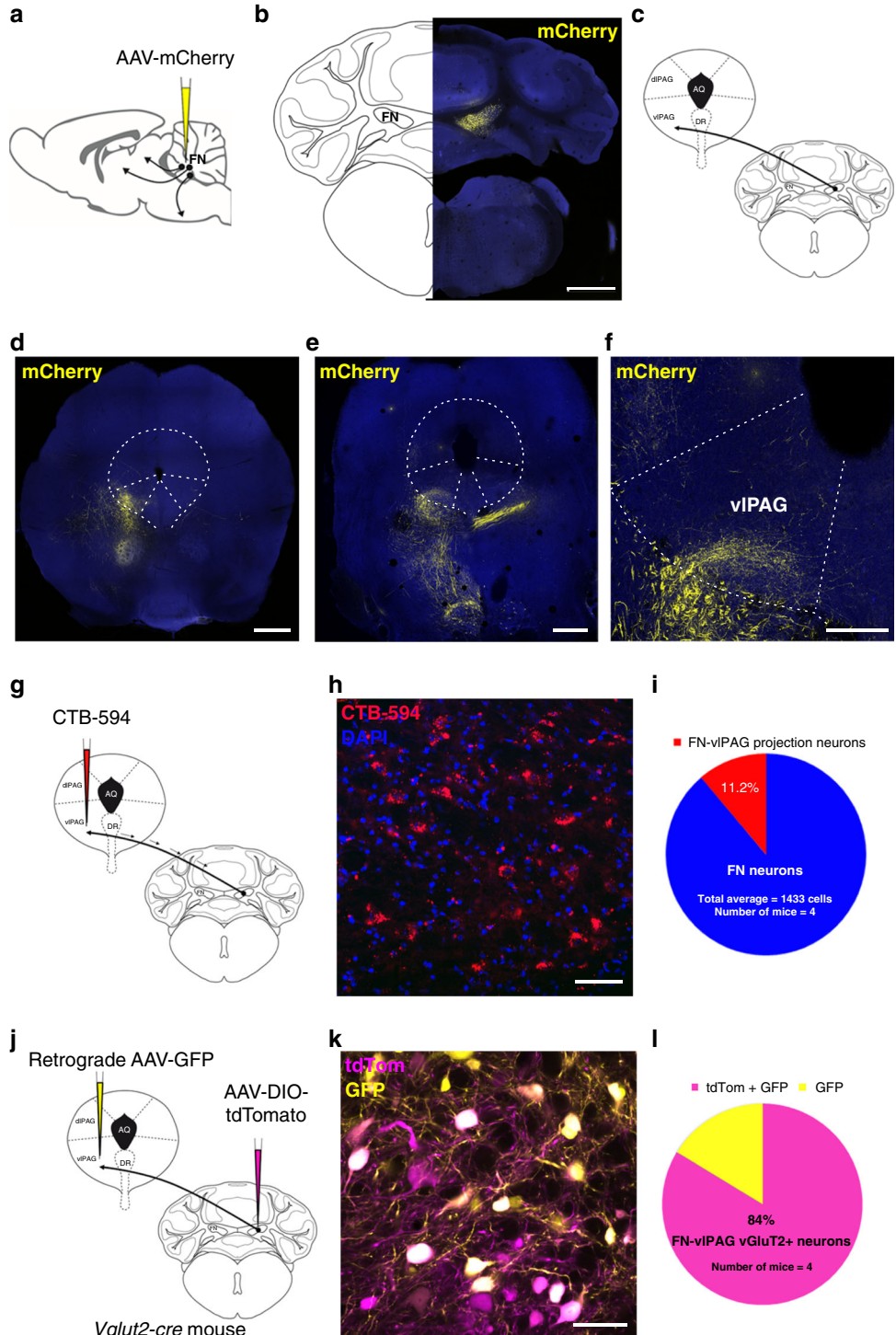

**Fig. 1 Cerebellum sends excitatory monosynaptic projections to vlPAG. a** Anterograde tracing strategy by AAV-mCherry injection in the FN.
**b** Expression of anterograde AAV-mCherry in the FN of the cerebellum (scale bar, 1 mm). **c** Schematic representation of anterograde tracing of AAV-mCherry from the FN into vlPAG. **d**, **e** Anterograde expression of mCherry in FN neuronal projections exhibiting varicosities in the vlPAG (scale bar, 500 μm), anterior (**d**) and medial (**e**) PAG sections. **f** Zoom-in from **d** (scale bar, 250 μm). **g** Retrograde labeling approach by injecting CTB-594 in the vlPAG. **h** Retrograde labeling of FN–vlPAG-projecting neurons with CTB-594, co-staining with DAPI (scale bar, 50 μm). **i** Quantification of retrograde labeled FN neurons projecting to the vlPAG ($n = 4$ mice, total average = 1433 cells). **j** Viral injection of retrograde AAV-GFP and anterograde AAV-DIO-tdTom into *vglut2-cre* mice strategy. **k** Co-expression of anterograde AAV-DIO-tdTom and retrograde AAV-GFP in vGluT2+ FN neurons projecting to vlPAG (scale bar, 50 μm). **l** Quantification of co-localizing retrograde AAV-GFP and AAV-DIO-tdTomato vGluT2+ neurons in the FN ($n = 4$ mice). Source data are provided as Source data file.

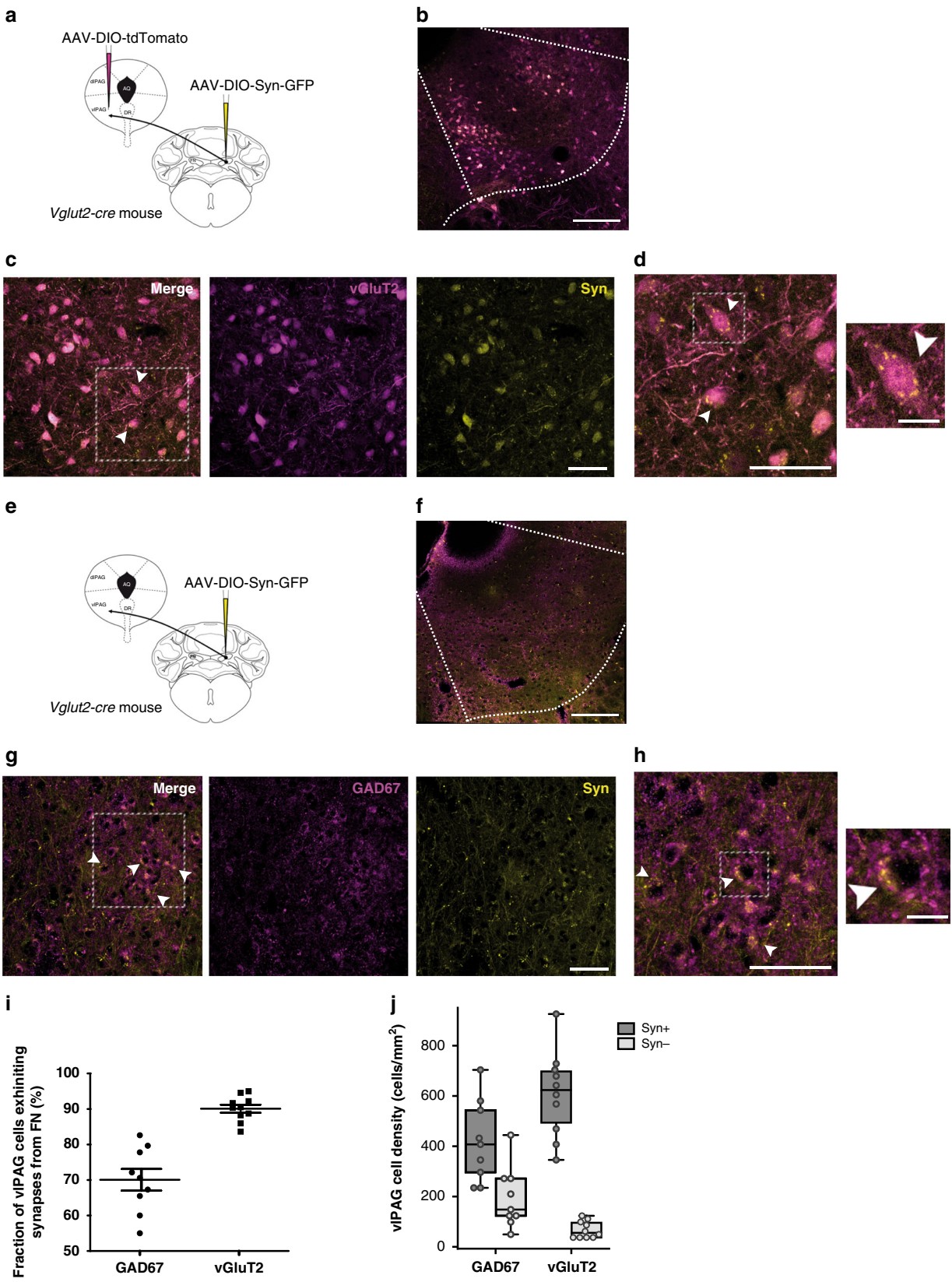

of response at the highest intensity range from 7 to 50 ms (Fig. 3f), suggesting the existence of direct and indirect excitation of the recorded vlPAG cells by FN input[26]. These results demonstrate the existence of a functional connectivity between the FN and the vlPAG.

**Cerebellar output to vlPAG controls fear memory formation.** To determine whether the FN–vlPAG plays a role in Pavlovian fear conditioning, we examined the effect of activating or inhibiting transiently these projections. First, we expressed excitatory or inhibitory DREADD receptors activated by CNO in FN

**Fig. 2 Glutamatergic FN input onto excitatory and inhibitory neurons in the vlPAG. a** Injection of cre-dependent AAV-DIO-Syn-GFP and AAV-DIO-tdTomato in vglut2-cre mice line strategy. **b** Expression of tdTomato in vGluT2+ neurons and Syn-GFP in FN terminals in the vlPAG (scale bar, 250 μm). **c** FN glutamatergic synapses contacting vGlut2+ neurons in vlPAG identified by expression of Syn-GFP (arrowheads, scale bar, 50 μm). Some of the somatic labeling likely reflects the high somatic tdTomato expression and cross talk detection with the GFP; intense synaptophysin labeling is best visualized in high-magnification panel. **d** High magnification from **c** (left panel, scale bar, 50 μm) and zoom-in of box area (right panel, scale bar, 12.5 μm). **e** Injection of cre-dependent AAV-DIO-Syn-GFP in vglut2-cre mice line strategy. **f** Immunostaining of GAD67 and Syn-GFP expression in FN terminals in the vlPAG (scale bar, 250 μm). **g** FN glutamatergic projections synapsing on vlPAG GABAergic neurons identified by co-staining of Syn-GFP and GAD67 in vlPAG (arrowheads, scale bar, 50 μm). **h** High-magnification from **g** (left panel, scale bar, 50 μm) and zoom-in of box area (right panel, scale bar, 12.5 μm). **i** quantification of GAD67 ($n = 9$ PAG slices) and vGluT2 ($n = 10$ PAG slices) cell fraction (%) exhibiting synaptophysin boutons on their cell bodies in the vlPAG. Data are presented as mean values ± SEM. **j** quantification of GAD67 ($n = 9$ PAG slices) and vGluT2 ($n = 10$ PAG slices) cell density (cells/mm$^2$) in the vlPAG showing synaptophysin boutons (Syn+) or not (Syn−). Data are presented as boxplots with dots representing data points, center line corresponds to the median of the distribution, the lower and the upper bounds of the box correspond to the 25th and 75th percentiles, respectively, and the whiskers correspond to 1.5 of the IQR (inter quartile ratio). Source data are provided as Source data file.

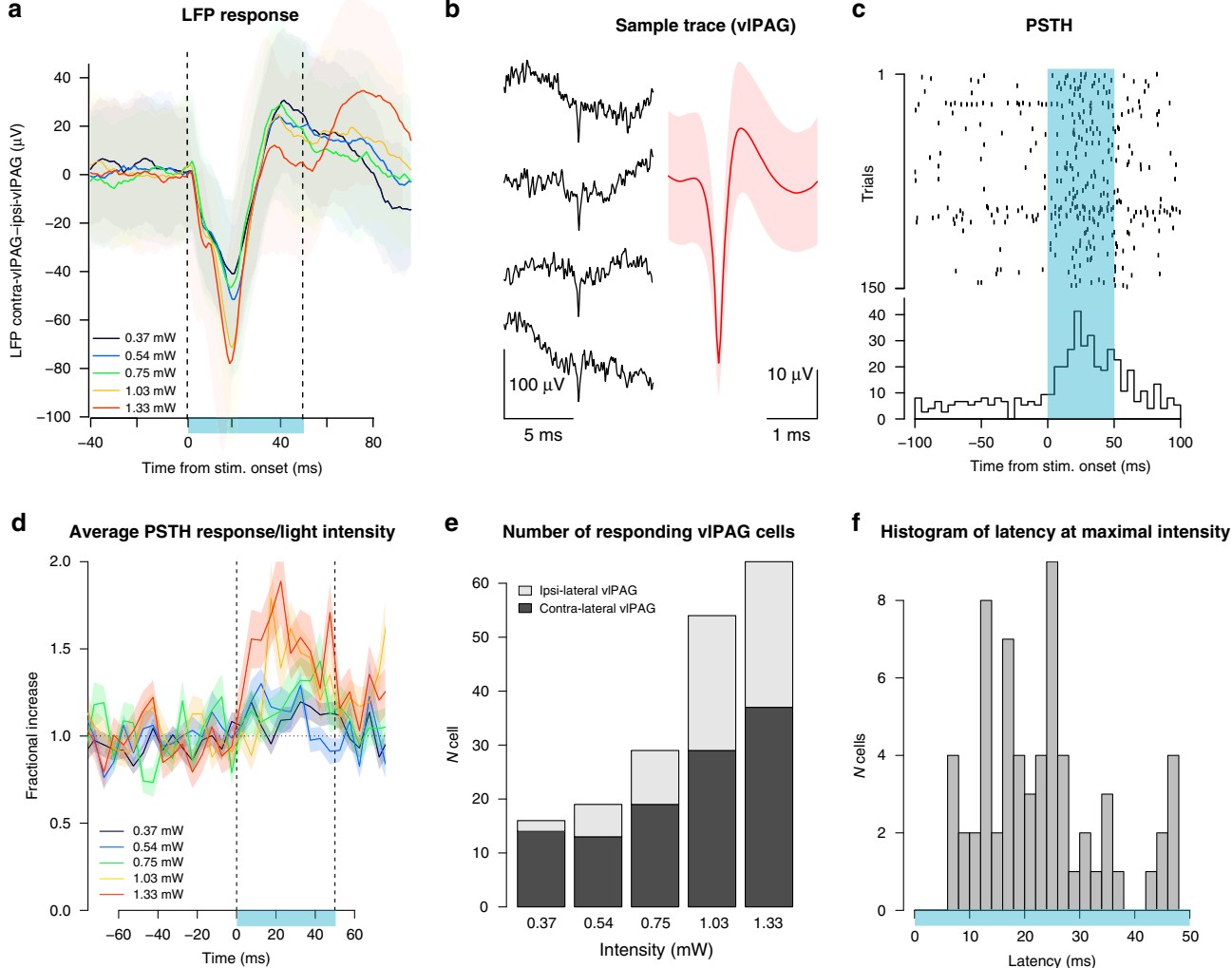

**Fig. 3 Optogenetic FN stimulation induces short-latency responses in vlPAG. a** LFP response in vlPAG to FN stimulation for five different light intensity showed with different colors. Error bands represent SEM. **b** Excerpt of unfiltered traces and spikes from a vlPAG unit; right: average filtered waveform for the neuron sorted on the channel; red line represent the mean value and shaded area represent the SD of the trace. **c** Raster plot for a cell recorded in vlPAG around the 50 ms period of FN stimulation. Blue area indicates periods with cerebellar optogenetic stimulation. **d** Light-dependent effect for the cells in the vlPAG that increased their firing during 50 ms light stimuli at each intensity bin = 5 ms ($n = 61$ out of 222 neurons from four mice with significant response at the highest intensity; significance is established with 2 ms bin, threshold used to identify significance in PSTHs is: 4.1 × SD). Error bands represent SEM. **e** Number of responding vlPAG cells (ipsi and contra-lateral) to FN stimulation ($n = 222$ neurons from four mice). **f** Latencies of responses in cells in vlPAG at highest intensity during the 50 ms of FN stimulation. Source data are provided as Source data file.

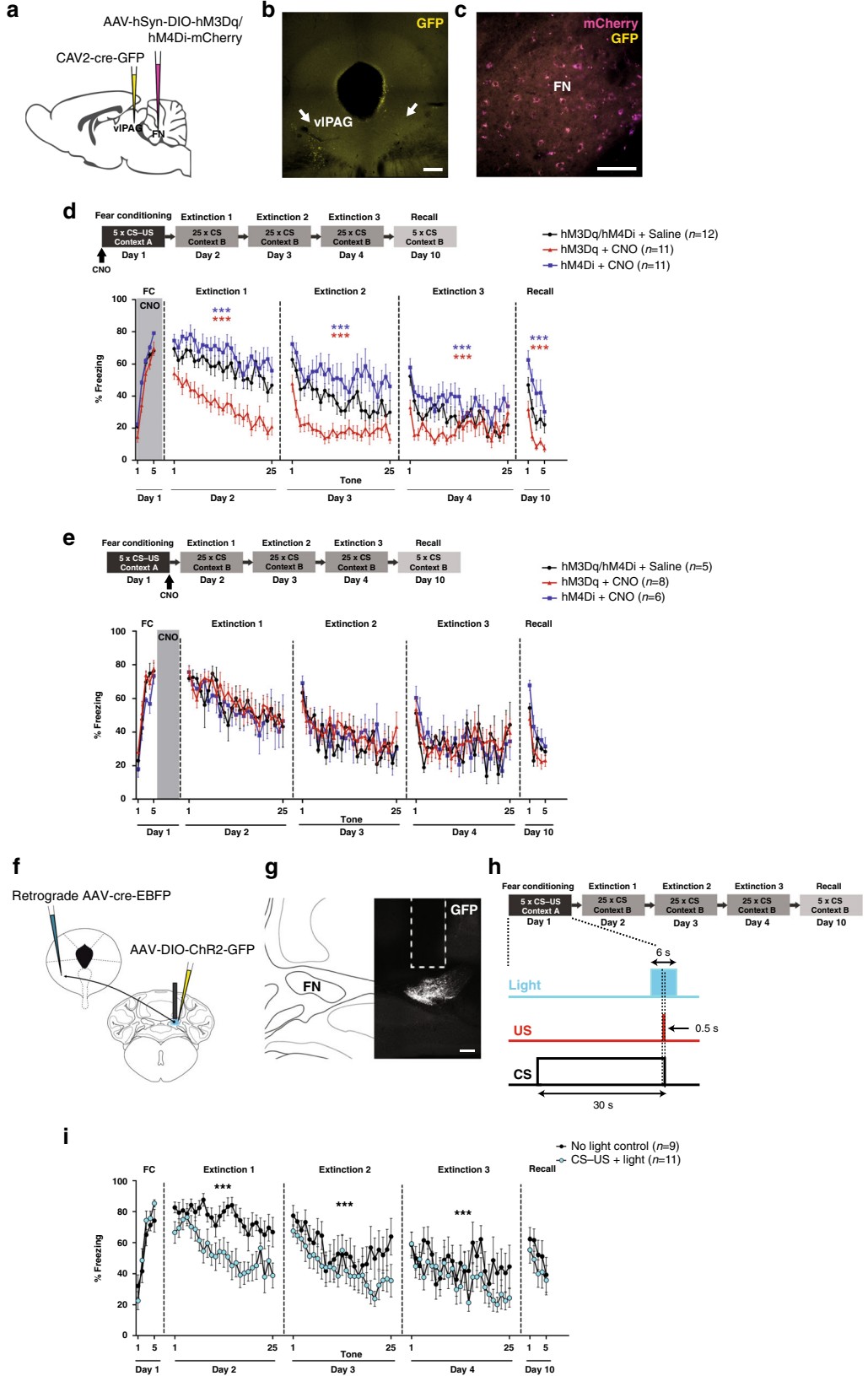

neurons that target the vlPAG, by infusing cre-dependent ante-rograde AAV-hM3Dq or -hM4Di in FN in combination with the infusion of retrograde CAV2-cre in the vlPAG (Fig. 4a–c). We performed a classical Pavlovian fear conditioning protocol, which consisted in five CS–US presentations, followed by three

extinctions sessions of 25 CS, and a final recall test of five CS presentations 1 week after to evaluate long-term maintenance of fear and extinction memories (Fig. 4d). Mice injected with saline and expressing the excitatory or the inhibitory DREADDs had similar freezing levels (Supplementary Fig. 3a), thus they were

**Fig. 4 FN–vlPAG projections control CS–US association and fear memory formation. a** Chemogenetic modulation of FN–vlPAG pathway, bilateral cre-dependent expression of excitatory or inhibitory DREADDs (hM3Dq or hM4Di, respectively) in the FN by retrograde CAV2-cre-GFP injection in the vlPAG. **b** Injection site of CAV2-cre-GFP in the vlPAG (arrows, scale bar, 200 μm). **c** Cre-dependent expression of DREADDs in the FN (scale bar, 100 μm). **d** Classical fear conditioning and extinction protocol (upper panel). FN–vlPAG pathway stimulation (hM3Dq + CNO, n = 11) or inhibition (hM4Di + CNO, n = 11) during fear conditioning (FC) exhibited similar freezing levels than the control group (hM3Dq/hM4Di + saline, n = 12; respectively, FC: $F_{(1,54)}$ = 0.0767, $P$ = 0.285; FC: $F_{(1,130)}$ = 1.15, $P$ = 0.285, two-way ANOVA). Stimulation of FN–vlPAG pathway during FC provoked a decrease in freezing levels during the extinction sessions (extinction 1: $F_{(1,546)}$ = 251.5, extinction 2: $F_{(1,546)}$ = 159.3, extinction 3: $F_{(1,546)}$ = 25.9, $P$ < 0.0001, two-way ANOVA), while inhibition induced an increase of freezing levels along the extinctions (extinction 1: $F_{(1,546)}$ = 27.5; extinction 2: $F_{(1,520)}$ = 46.5, extinction 3: $F_{(1,546)}$ = 25.3, $P$ < 0.0001, two-way ANOVA), compared to the control group. During the recall, stimulated mice expressed less fear than the control group (recall: $F_{(1,120)}$ = 0.36, $P$ < 0.0001, two-way ANOVA), while inhibited mice expressed higher levels of freezing (recall: $F_{(1,102)}$ = 16.1, $P$ < 0.0001, two-way ANOVA). **e** Stimulation (hM3Dq + CNO, n = 8) and inhibition (hM4Di + CNO, n = 6) of FN–vlPAG pathway during consolidation phase post-FC had not effect on freezing behavior (hM3Dq/hM4Di + saline, n = 5; stimulation: extinction 1: $F_{(1,275)}$ = 0.57, $P$ = 0.347, extinction 2: $F_{(1,250)}$ = 3.50, $P$ = 0.624, extinction 3: $F_{(1,250)}$ = 0.37, $P$ = 0.543; inhibition: extinction 1: $F_{(1,200)}$ = 0.98, $P$ = 0.324, extinction 2: $F_{(1,200)}$ = 1.38, $P$ = 0.241, extinction 3: $F_{(1,200)}$ = 0.09, $P$ = 0.770; two-way ANOVA). **f** Optogenetic stimulation of FN–vlPAG inputs by cre-dependent expression of ChR2-GFP in FN and retrograde cre in vlPAG. **g** ChR2 expression in FN–vlPAG-projecting neurons (scale bar, 200 μm). **h** Optogenetic stimulation protocol of FN–vlPAG pathway during CS–US presentation in fear conditioning. Light pulses were delivered every CS–US presentation. **i** Mice under light stimulation during CS–US pairing (CS-US + light, n = 11) exhibited an enhanced extinction of fear response compared to the control mice (no light control, n = 9; extinction 1: $F_{(1,450)}$ = 104.1, extinction 2: $F_{(1,422)}$ = 18.8, extinction 3: $F_{(1,455)}$ = 13.4, $P$ < 0.0001, two-way ANOVA). Lines represent means ± SEM. ***$P$ < 0.001. Source data are provided as Source data file.

grouped together in the control (injected with saline) group. In addition, we checked that CNO had no effect per se on the freezing behavior by performing the fear and extinction protocols on control sham mice. Mice exposed to CNO indeed exhibited the same freezing levels than the control mice, which received saline (Supplementary Fig. 3b). Moreover, chemogenetic activation or inhibition of FN–vlPAG pathway had not effect on the immobility state of the mice in an open field compared to the saline group (Supplementary Fig. 3c), neither on the freezing levels in the context A before fear conditioning (Supplementary Fig. 3d) or in the context B before extinction training (Supplementary Fig. 3e). These results indicate that FN–vlPAG pathway does not affect the immobility or freezing behavior.

We then analyzed the impact of the modulation of the FN–vlPAG pathway on fear conditioning. For this purpose, we activated or inhibited DREADD-expressing FN–vlPAG-projecting neurons with CNO during the acquisition phase. Under these conditions, mice exhibited similar fear response than the control group. Strikingly, during the first extinction training the following day, mice in which the FN–vlPAG pathway had been activated during fear conditioning, exhibited a drastic drop in freezing behavior along the CS presentations, reaching basal freezing levels already at the end of the first extinction session (Fig. 4d). During the second extinction training, this group of mice succeeded to retrieve the fear memory at the first CS presentation, but they reached basal levels of freezing at the third CS presentation, indicating a faster extinction than in the control mice. In contrast, mice in which the FN–vlPAG pathway had been inhibited presented a stronger freezing behavior throughout the first extinction training, as well as for the second and third extinction sessions, compared to the control mice (Fig. 4d). Moreover, when we evaluated the long-term maintenance of the fear and extinction memories by a recall session of five CS presentations 1 week after, mice that underwent FN–vlPAG activation during fear conditioning exhibited a lower recall of fear memory than the control mice, while mice that underwent FN–vlPAG inhibition exhibited an increased fear memory compared to the control mice. These results indicate that FN–vlPAG projections contribute to the formation of conditioned fear memories, in which the activation of this pathway during the fear conditioning decreases the strength of the fear memory formation, while the inhibition of this pathway increases it. Consequently, the extinction of fear response during the extinction session is faster or slower, respectively.

In addition, we found an increase of c-Fos+ vlPAG neurons following fear conditioning when FN–vlPAG pathway was chemogenetically activated, compared to the control (Supplementary Fig. 4a, b), confirming that FN–vlPAG glutamatergic projections induce an increase in neuronal recruitment when it is stimulated during fear conditioning.

Since the systemic CNO administration exerts an effect on neuronal activity for hours[34–36], we wondered whether the bidirectional control of FN–vlPAG pathway on fear memory occurs specifically during the acquisition or during the consolidation phase that immediately followed[37,38]. To answer this question, we compared the effect of activating and inhibiting the FN–vlPAG pathway starting from the acquisition phase or from the consolidation phase. Then, we evaluated the fear response during the following recall and extinction sessions. We surprisingly found that activating and inhibiting the FN–vlPAG pathway only during the consolidation phase of fear memory had no effect on the strength of the fear memory (Fig. 4e). Mice with FN–vlPAG activation or inhibition exhibited similar levels of fear response than the control group, during the recall and along the extinctions sessions. Taken together, these results indicate that the bidirectional control of the FN–vlPAG pathway of the fear memory occurs during the acquisition phase of the fear memory, in which the association of the CS–US is taking place.

**FN–vlPAG pathway activation at CS–US reduces fear learning.** Furthermore, to evaluate whether the state of the FN–vlPAG projections regulates the fear learning at the time of the CS–US association[39,40], we optogenetically stimulated the FN during the CS–US presentation (from the last 3 s of the CS until 3 s post CS–US, Fig. 4f–h). We found that light stimulation during CS–US association is sufficient to induce a significant enhanced extinction of the fear response, during the extinction sessions compared to the control group (Fig. 4i). Moreover, the light stimulation had not effect on the immobility state in an open field (Supplementary Fig. 5b) or on the freezing behavior, during the fear acquisition phase (Fig. 4i). Therefore, these results indicate that the cerebellum exerts a control over the fear memory formation through the FN–vlPAG pathway during the association between the CS and US, without affecting the fear behavior during conditioning.

Interestingly, mice optogenetically stimulated randomly within the intertrial interval (ITI) exhibited a reduced freezing during fear conditioning, suggesting a weaker fear during learning (Supplementary Fig. 5c, d), followed by a lower fear response

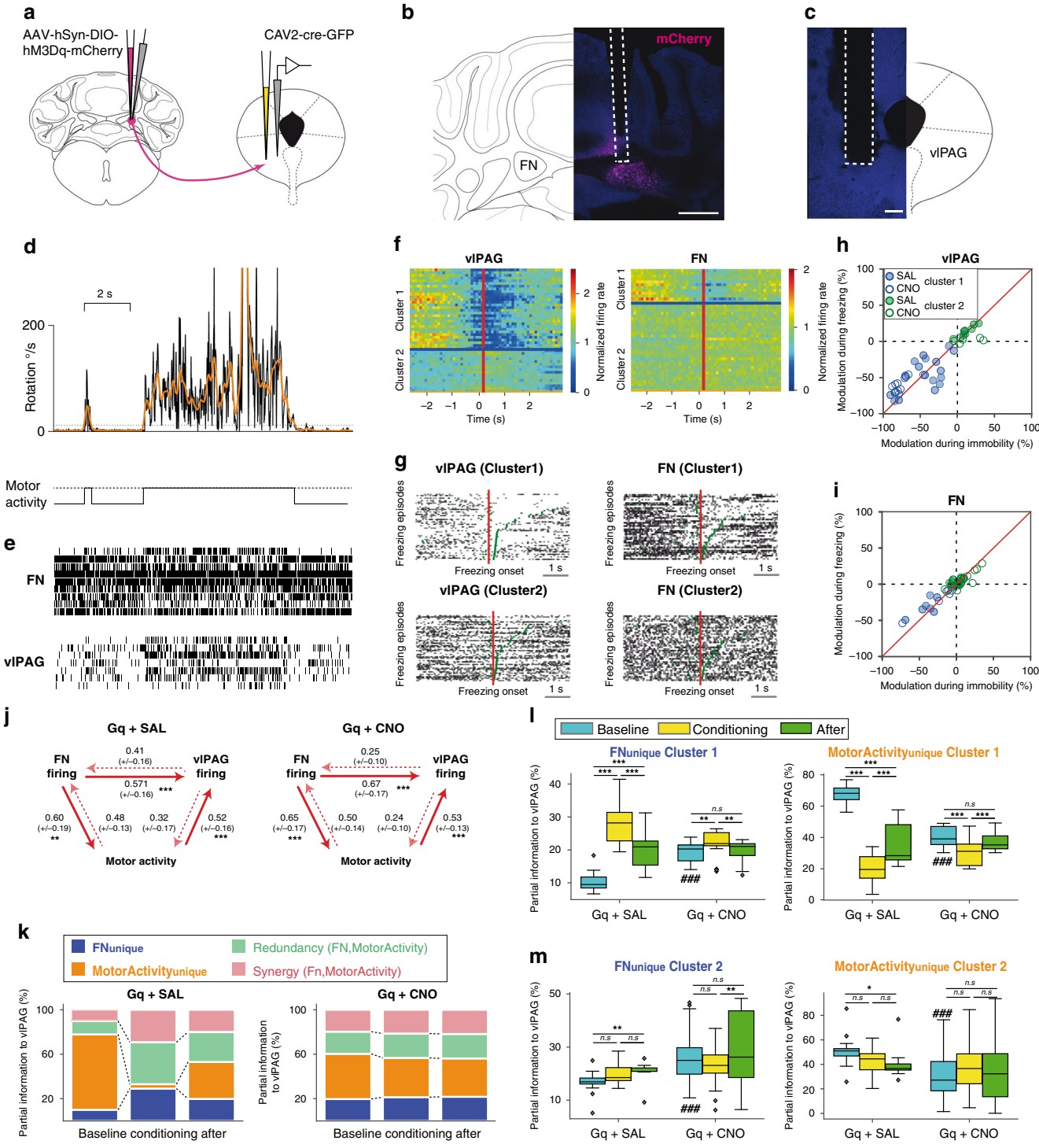

during retrieval of fear memory, and along the first and second extinction sessions, compared to the mice that were not stimulated. Thus, this suggests that FN–vlPAG projections could also contribute to US-induced fear extrinsically from the CS–US association coding.

**Cellular FN–vlPAG interaction during fear learning**. Therefore, to assess the neuronal correlates of fear learning in FN and vlPAG, we recorded extracellular activity during the conditioning. In order to relate the firing activity in the FN and vlPAG to the mouse immobility, we combined the cellular recordings with

measurements of the mouse movements, using an inertial sensor attached to the head of mice expressing an excitatory DREADD in the FN–PAG neurons receiving saline (Gq + SAL group) or CNO (Gq + CNO group; Fig. 5a–e).

To examine the link between cellular activity and freezing/immobility, we examined the modulation of firing around the onset of immobility in vlPAG and FN neurons (Fig. 5f). Two classes of association with immobility were observed in the vlPAG and in the FN: a set of cells ("cluster 1") exhibited a strong reduction of firing at the onset of immobility (Gq + SAL group, PAG: 17 cells, FN: 7 cells from 4 mice; Gq + CNO group, PAG: 13, FN: 5 cells from 3 mice), while a second set ("cluster 2")

**Fig. 5 Firing of FN and vlPAG neurons exhibit strong modulations as a function of motor activity and increased directional FN to vlPAG interactions during conditioning. a** Schematic of the experiment. **b** Histology of electrode position in the vlPAG (scale bar, 200 μm). **c** hM3Dq expression in FN–vlPAG-projecting neurons and electrode position (scale bar, 500 μm). **d** Top: instantaneous head rotation (black line) and smoothed rotations (orange line). Dashed line: threshold of immobility (12 deg s$^{-1}$). Bottom: binarized motor activity. **e** Raster of units recorded simultaneously in the FN and vlPAG, same time period as **c**. **f–j** Modulation of cell firing in the vlPAG and FN in relation with immobility **f**, PSTH of the vlPAG (left) and FN (right) neurons around the onset of immobility events; each line correspond to a different cell. Cells are grouped in cluster 1 and cluster 2 determined by using a $K$-means algorithm on the PSTHs. **g** Example rasters of clusters 1 and 2 cells in vlPAG and FN. Each plot corresponds to a single cell and each line to a different immobility episode. The rasters are aligned around the onset of immobility episodes and ordered by decreasing duration of the immobility episodes. The end of the immobility episode (and its preceding episode) is indicated in green. **h** Comparison of the change of vlPAG firing between mobility and immobility during baseline and conditioning (when immobility largely reflects freezing). The red line indicates an identical modulation in immobility and in freezing. **i** Same as **h** for FN neurons. **j** Pairwise information transfer between the vlPAG, FN firing, and binarized motor activity ("motor activity"); bin: 10 ms, lag: 1 bin. The information transfer is averaged over all FN–vlPAG pairs, and all FN and vlPAG unit and motor activity. Left: unit pairs Gq + SAL group. Right: units from the Gq + CNO group. Values correspond to mean ± SEM of the entropy transfer (in bits), **$P < 0.01$, ***$P < 0.001$, Mann–Whitney. **k–m** Partial information decomposition using FN firing and motor activity as sources, and vlPAG firing as target (bin: 10 ms, lag: 1 bin). **k** Example of evolution of the proportion of information transfer components from motor activity, and one FN neuron to a vlPAG neuron during the 5 min before (baseline), during (conditioning), and after (after) conditioning. The information transfer is computed at a 10 ms latency. **l** Evolution of the proportion of unique contribution from cluster one FN (left) and motor activity (right) to vlPAG firing. Tukey's post hoc comparisons: *$P < 0.05$, **$P < 0.01$, ***$P < 0.001$. Comparison between Gq + SAL and Gq + CNO, Mann–Whitney: ###$P < 0.001$, ##$P < 0.01$. **m** Same as **l** for cluster 2. **l, m** Boxplots the center line corresponds to the median of the distribution, the lower and the upper bounds of the box correspond to the 25th and 75th percentiles, respectively, and the whiskers correspond to 1.5 of the IQR (inter quartile ratio), the dots correspond to outliers. **j, l, m** statistics are detailed in Supplementary Table 1. Source data are provided as Source data file.

only exhibited mild modulations (Gq + SAL group: PAG: 8, FN: 11 cells from 4 mice; Gq + CNO group: PAG: 8, FN: 20 cells from 3 mice). We then examined the modulation of the cells in response to the CS and US (Supplementary Fig. 6). A number of cells exhibited responses after the onset of to the CS and after the US; however, these cells were observed in the cluster 1 group, and are thus strongly modulated by motor state, which switches to active both at the CS and US onsets. Although the change in firing at CS/US onset may correspond to fear prediction errors[41,42], we cannot exclude that the change motor state is responsible for the changes of firing at CS and US onset. The reduction of firing was usually observed throughout the period of immobility (Fig. 5g). To test whether the reduction of firing was correlated to immobility or to freezing, we compared the firing modulation between immobility and active state during the 5 min preceding conditioning and during conditioning, when immobility reflects freezing. We found a strong correspondence between the modulation observed in relation to immobility before and after conditioning, no cell exhibiting a clear modulation unique to freezing neither in the vlPAG nor in the FN (Fig. 5h, i); thus, although this result does not rule out the presence of freezing-related modulations of firing in the FN and vlPAG, the modulations observed seem primarily to reflect a motor activity.

To look at the interdependencies between FN and vlPAG firing, and take into account immobility, we turned to information theory approaches[43], which provide unbiased estimates of the relationship, whether linear or nonlinear, between time series. We first used the transfer of entropy, a specific case of conditional mutual information, to examine pairwise, temporally ordered, transmission of information between FN and PAG firing (Fig. 5j). Consistent with the observation of direct connections from the FN to the vlPAG reported above, we observed stronger transfer of entropy from the FN to vlPAG than the reverse. However, as suggested by the modulations reported in the Fig. 5f–i, there was also transfer of entropy between the motor activity and the FN and vlPAG firing, with a stronger transfer from the FN toward the motor state and motor state toward the vlPAG firing than the reverse (Fig. 5j). Therefore, the vlPAG units recorded in our study seem more strongly influenced by the FN and the motor activity than the reverse.

To disentangle the contributions of FN and motor state on the vlPAG firing, we used a partial information decomposition (PID) which breaks down the interdependency between two sources and

one target into four components: the unique contribution of each source, and two terms of interactions ("redundancy" and "synergy", respectively, corresponding to the transfer on the target of the linear and nonlinear interactions of the sources). We then determined these components during 5 min before ("baseline"), during ("conditioning"), and 5 min after ("after") conditioning using the (binarized) mobility signal ("motor activity"), and the firing from single FN neurons as sources and the firing of single PAG neurons as targets. The amplitude of information transfer is dependent on the variability of the signals, and thus can hardly be compared between conditions; however, the proportion of the components of information transfer can be compared across conditions (Fig. 5k).

In the baseline condition, the unique contribution of the FN to vlPAG firing is only a small fraction of the total contribution of FN and motor activity, but is significantly lower in the control group than in the Gq + CNO group (Fig. 5l, m); this indicates that the excitation of the FN by the CNO increases its relative contribution to PAG firing.

For cluster 1 units, the proportion of unique contribution of FN neurons to vlPAG firing increases relative to baseline during conditioning in the Gq + SAL group (Fig. 5l, m), while the proportion of information transfer carried by the motor state alone (MotorActivity$_{unique}$) exhibited the reciprocal evolution; cluster 1 units from the Gq + CNO group exhibited similar, but smaller modulation (significant ANOVA interaction stage × treatment; Supplementary Table 1). Cluster 2 units exhibited less significant modulations between the conditions in the Gq + SAL group, suggesting a weaker contribution of these units. This indicates that a transmission of information from the FN to the vlPAG increases during conditioning when learning is taking place, but exhibit weaker evolutions when the FN is excited by the DREADD.

**Fastigial inputs to the vlPAG modulate extinction learning.** We also examined whether the cerebellum plays a role in fear extinction. For this purpose, mice expressing hM3Dq or hM4Di in the vlPAG-projecting FN neurons were exposed to CNO, during the first and second extinction sessions (Fig. 6a, b). Fear-conditioned mice were subjected to the first recall and extinction training the following day under CNO effect. During the first CS presentations, the different groups of mice exhibited the same levels of freezing, suggesting that FN–vlPAG neurons do not

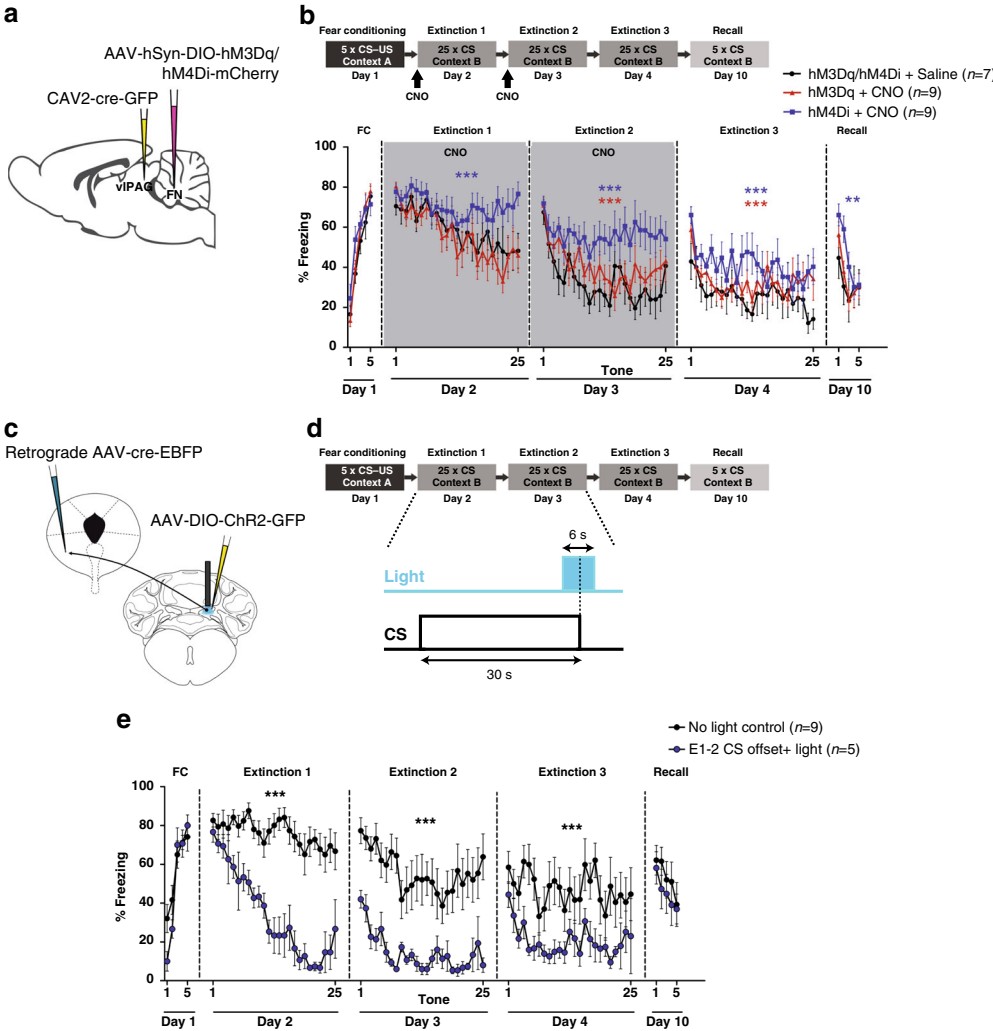

**Fig. 6 Involvement of FN–vlPAG pathway in fear extinction learning. a** Chemogenetic strategy to stimulate or inhibit the activity of FN–vlPAG-projecting neurons by bilateral cre-dependent expression of excitatory or inhibitory DREADDs (hM3Dq or hM4Di, respectively) in the FN in combination, with retrograde CAV2-cre-GFP injection in the vlPAG. **b** Stimulation and inhibition of FN–vlPAG pathway during first and second extinction sessions. Inhibition of FN–vlPAG pathway induced higher levels of freezing behavior during the extinction sessions under effect of CNO, and along the third extinction session and recall test without CNO, compared to the control group (hM3Dq/hM4Di + saline, $n = 7$; hM4Di + CNO, $n = 9$; extinction 1: $F_{(1,364)} = 37.9$, extinction 2: $F_{(1,364)} = 96.9$, extinction 3: $F_{(1,364)} = 44.2$, $P < 0.0001$, two-way ANOVA; recall: $F_{(1,78)} = 10.5$, $P = 0.0017$, two-way ANOVA). Stimulation of this pathway during the extinctions also induced an increase of freezing behavior during the second and third extinction sessions (hM3Dq/hM4Di + saline, $n = 7$; hM3Dq + CNO, $n = 9$; extinction 2: $F_{(1,364)} = 12.7$, extinction 3: $F_{(1,364)} = 12.0$, $P < 0.0001$, two-way ANOVA), but not during the recall (recall: $F_{(1,78)} = 0.87$, $P = 0.353$, two-way ANOVA). **c** Optogenetic stimulation of FN–vlPAG projections expressing cre-dependent ChR2-GFP strategy. **d** Optogenetic stimulation protocol of FN–vlPAG-projecting neurons during CS offset in extinction sessions 1 and 2 (E1-2). **e** Freezing levels during fear conditioning and extinction sessions in control (no light control, $n = 9$) and FN–vlPAG stimulated mice (E1-2 CS offset + light, $n = 5$). Mice that received light stimulation during CS offset exhibited a enhanced curve of extinction learning during the first and second extinctions, and lower fear response during third extinction in absence of light stimulation, compared to the mice that were not stimulated (extinction 1: $F_{(1,306)} = 372.3$, extinction 2: $F_{(1,281)} = 212.7$, extinction 3: $F_{(1,306)} = 65.04$, $P < 0.0001$, two-way ANOVA), but there was not significant difference during the recall (recall: $F_{(1,61)} = 1.32$, $P = 0.254$, two-way ANOVA). Lines represent means ± SEM. **$P < 0.01$, ***$P < 0.001$, two-way ANOVA. Lines represent means ± SEM. Source data are provided as Source data file.

affect the fear memory retrieval (Fig. 6b). In addition, control sham mice exposed to CNO during the first and second extinction showed similar freezing levels compared to the control group (Supplementary Fig. 3b), indicating that CNO has not effect on fear extinction in absence of DREADD receptors. During the first extinction session, mice under FN–vlPAG activation did not exhibit differences in the freezing levels compared to the control group. However, mice under FN–vlPAG inhibition exhibited impairment in the extinction of the fear response during the first and second extinction training (Fig. 6b). Moreover, at the third extinction session, without exposure to CNO, these mice still exhibited higher levels of freezing compared to the control

group, suggesting that FN–vlPAG neurons could also control the extinction learning.

Moreover, to evaluate whether similar to the role on the CS–US association in fear conditioning, this FN inputs to vlPAG might also have a contribution to the negative fear prediction error ("false alarm") or new association learning in extinction, we performed the optogenetic stimulation of FN–vlPAG projections during the CS offset along the first two extinction sessions (Fig. 6c, d). Interestingly, we found that the light stimulation of FN–vlPAG ChR2-expressing neurons at CS offset enhanced significantly the extinction learning from the first extinction session; this was not due just to the decrease on fear expression

but to learning, which is evidenced by the lower fear levels in the recall during the second extinction (Fig. 6e). This effect was also sustained in the absence of light stimulation during the third extinction. Interestingly, contrarily to fear learning, the sign of the effects on fear extinction of excitatory DREADD and of the optogenetic stimulation are different. This suggests that the effect on extinction depends on the temporal pattern of activation of the FN–vlPAG neurons. Overall, these results indicate that stimulation of the FN–vlPAG pathway may considerably accelerate the fear extinction learning.

Taken together, our results indicate that FN inputs to the vlPAG are able to control in a bidirectional way the strength of fear memory encoding, reinforcing the CS–US association when this pathway is inhibited or diminishing the CS–US association when it is activated. In addition, FN–vlPAG neurons may also exert a robust modulation of extinction learning.

**FN–vlPAG projections send collaterals to other areas**. We analyzed whether FN projections to vlPAG have axon collaterals targeting other areas in the brain, through the expression of retrograde AAV-cre-EBFP in vlPAG neurons in combination with cre-dependent Syn-GFP in FN, to evaluate the presence of synaptic boutons, together with cre-dependent tdTomato, to trace the axonal projections (Supplementary Fig. 7a). We were able to determine few axon collaterals from the FN–vlPAG-projecting neurons (Supplementary Fig. 7b, c) to different nuclei in the thalamus, such as the thalamic parafascicular nucleus (PF, Supplementary Fig. 7d), the mediodorsal nucleus (Supplementary Fig. 7e), the ventromedial nucleus (Supplementary Fig. 7e), and the ventral-anterior lateral nucleus (Supplementary Fig. 7f). Thus, as FN–vlPAG pathway, these axon collaterals could also impact on fear memory.

**Neuroanatomical link between the cerebellum and the amygdala**. Since FN–vlPAG neurons have also some axon collaterals to thalamic nuclei, we studied whether any of these nuclei have a link with the BLA, which is a key brain structure for fear memory acquisition and consolidation. Therefore, in order to evaluate a possible neuroanatomical connection between the cerebellum and the BLA, we combined anterograde tracing from the FN with retrograde tracing from the BLA (Fig. 7a–c). This analysis revealed BLA-projecting neurons in the area of the thalamic PF receiving neuronal projections from FN (Fig. 7d, e), suggesting that PF could be a relay area between the cerebellum and the BLA.

**FN projections to PF does not modulate fear memories**. To test if these FN projections to the PF play a role on fear learning, we examined the effect of activating or inhibiting this pathway during fear conditioning in mice expressing cre-dependent anterograde AAV-hM3Dq or -hH4MD injected in FN neurons, and retrograde CAV2-cre injected in the PF (Fig. 8a–c). Chemogenetic modulation of FN–PF projections did not induce significant differences in freezing behavior during fear conditioning or during extinction compared to the control mice (Fig. 8d). In addition, either activation or inhibition had not effect on the basal freezing levels in the context A before fear conditioning (Fig. 8e). Therefore, these results indicate that the FN–PF neurons do not contribute to fear memory formation, despite the proximity of their terminals with neurons projecting to the BLA.

**FN–PF projections contribute to anxiety-like behavior**. The neuronal circuits for fear and anxiety are overlapping and closely related[7,44]. Therefore, we evaluated if the FN–vlPAG or the FN–PF pathways contribute to anxiety behavior in three different anxiety tasks: open field, elevated plus maze and dark–light box. We found no difference in anxiety parameters that indicate generalized anxiety-like behavior, neither when FN–vlPAG pathway was activated or inhibited (Supplementary Fig. 8a), nor in the sham group exposed to CNO (Supplementary Fig. 8c). On the other hand, mice under activation of FN–PF neurons exhibited a decrease in the frequency of entries in the open arms in the EPM, and decreased frequency of entries into the light zone in the light–dark box test, indicating an anxiogenic effect induced by the activation of these neurons (Supplementary Fig. 8b). Therefore, these results suggest that FN–PF projections modulate anxiety levels, while FN–vlPAG does not affect anxiety, but has a specific role on fear conditioning.

Therefore, despite the existence of collaterals of FN–vlPAG in the PF, the chemogenetic manipulations of FN–vlPAG and FN–PF neurons produce distinct, nonoverlapping effects, the former impacting fear learning and the latter anxiety.

**FN inputs to vlPAG and PF do not affect pain sensitivity**. PAG is involved in processing pain information from the periphery[13]. Since the activation or inhibition of FN–vlPAG neurons during CS–US presentations respectively diminished or reinforced fear memory, we tested if that could result from an alteration on the sensitivity to painful stimuli by hot plate and tail-immersion tests. No differences were found in the hot plate or the tail-immersion sensitivities for FN–vlPAG (Supplementary Fig. 8d, e) or FN–PF projections (Supplementary Fig. 8f, g), indicating that those inputs do not modulate the sensitivity to painful stimuli. Therefore, alteration in sensitivity is not responsible for the effect of FN–vlPAG inputs on fear memories formation and expression.

## Discussion

Despite many clinical and anatomical evidence indicating an involvement of the cerebellum in emotional functions[19], the nature of its contribution remains unclear. Here, we describe a link between the cerebellum and one element of the fear circuitry, the vlPAG, through which it exerts a bidirectional control over fear learning during the association of a cue and aversive events. Our experiments reveal that glutamatergic neurons in the FN send projections to the vlPAG where they form synapses onto glutamatergic and GABAergic neurons. The chemogenetic or optogenetic activation of this pathway during fear conditioning increases the activity of vlPAG neurons, but yields reduced fear expression during the recall and extinction. In contrast, the inhibition of this pathway during fear conditioning results in a higher expression of conditioned fear and a slower extinction learning. Thus, our experiments are consistent with a participation of the cerebellum to the function of the vlPAG in fear learning and memory (Fig. 9).

The PAG has long been implicated as the organizer of behavioral components of the response to threat. The most characterized function of the vlPAG is its direct control of freezing via glutamatergic projections to premotor targets in the magnocellular nucleus of the medulla; this pathway is recruited in innate freezing, and in cued- or context-driven freezing respectively by GABAergic afferents from CEA[10], and glutamatergic afferents from the mPFC[6]. The cerebellum may also participate to freezing expression. Indeed, the vlPAG has been shown to entrain (via an unknown pathway) the climbing fiber in the lobule 8 of the rat cerebellum, a region found to reduce both innate and cued-freezing expression if lesioned[24], and recent in vitro work indicate functional connections to freezing-related neurons[26]. This modulation on freezing expression is unlikely supported by the cerebellum–vlPAG projections targeted in our experiments, since the chemogenetic or optogenetic stimulation of the FN–vlPAG

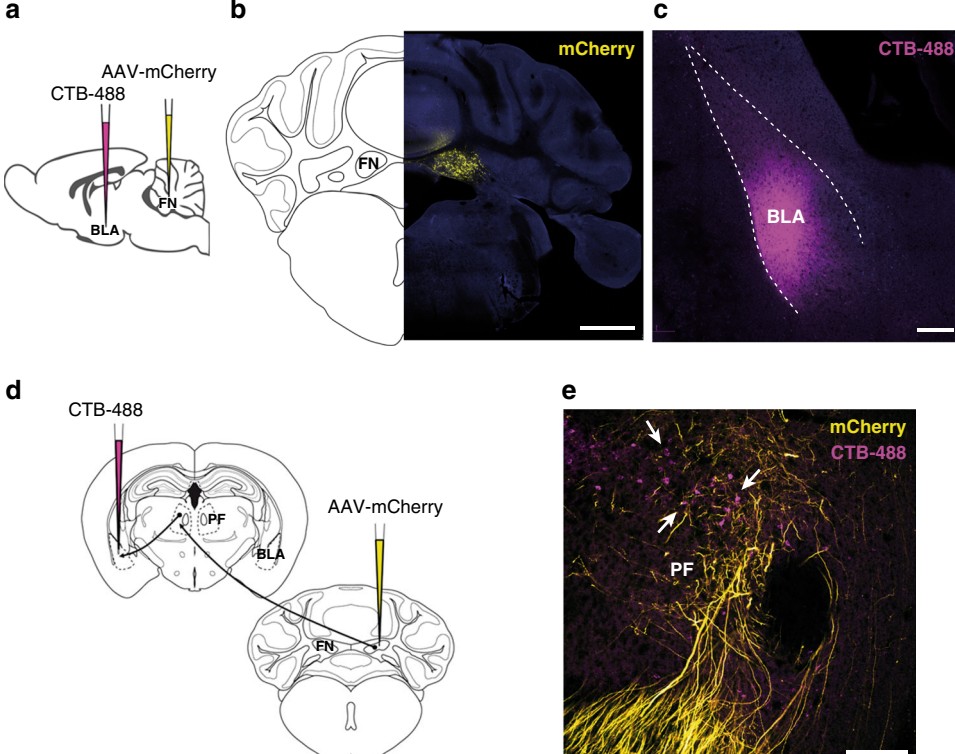

**Fig. 7 Cerebellar link to the BLA through thalamic PF. a** BLA injection of retrograde CTB-488 in combination with anterograde AAV-mCherry in the FN strategy. **b** Expression of anterograde AAV-mCherry in the FN (scale bar, 1 mm). **c** Infusion of CTB-488 in the BLA (scale bar, 200 μm). **d** Retrograde tracing from BLA combined with anterograde tracing from the FN converged at the thalamic PF nucleus. **e** PF neurons projecting to the BLA (arrows) surrounded by fibers from FN (scale bar, 200 μm).

did not affect the freezing evoked by the aversive stimulus during the presentation of the US, and failed to modify the cued-freezing at the time of recall (i.e., beginning of the extinction training). However, we found a small reduction of freezing when the FN–vlPAG neurons were stimulated between the CS–US pairing during the fear acquisition. Therefore, our results unlikely result from an entrainment by FN–vlPAG of the vlPAG neurons projecting to the magnocellular medulla responsible for freezing, or to their afferent inhibitory neurons. Thus, the effects revealed in our work would not result from the regulation of the motor control of freezing behavior, but from a modulation of fear memories.

The changes in the conditioned fear response observed during extinction following the modulation of FN–vlPAG neurons during fear conditioning, shall indeed result from a change in the learning of the CS–US association. A trivial explanation could be that the FN–vlPAG pathway modulates the sensory perception of the US at the time of fear encoding. Although excitatory and inhibitory neurons in the vlPAG exert an antinociceptive effect[10,11], we found that chemogenetic manipulations failed to change the responses to pain in the tail-immersion and hot plate tests. This indicates that the changes in fear learning were not due to an effect of the FN–vlPAG directly on the antinociceptive circuit of the vlPAG, also consistent with the similar fear levels (freezing) exhibited by the animals induced by the US. An alternative hypothesis to explain the changes in fear learning observed during the extinction is an alteration of fear consolidation[28]; however, the modulation of the FN–vlPAG pathway during consolidation phase after fear conditioning had not effect on the retrieval of the conditioned fear response in the following days. Moreover, the optogenetic stimulation of the pathway only at the time of the CS–US presentation was sufficient to induce a

robust decrease in the conditioned fear expression in the following days, consistent with a modulation on the encoding of the CS–US association and a limited contribution of the FN–vlPAG pathway to the consolidation process.

At odds with the classical view of the PAG as a relay mediating emotional responses, increasing evidence points toward a bidirectional contribution of vlPAG neurons to fear learning: the chemogenetic activation of vlPAG during learning subsequently reduces the cued feeding suppression indicating a reduced cued fear[45] (consistent with the chemogenetic and optogenetic findings of the present work), while the pharmacological inhibition of vlPAG before CS–US presentation may prevent learning of freezing responses[46]. Some of the effect of the vlPAG on fear learning may involve a cued-antinociception triggered by an amygdala–vlPAG pathway[12], although we have not observed a change in sensitivity to aversive stimuli in our chemogenetic experiments. However, the optogenetic inhibition of vlPAG has been found to inhibit the acquisition of conditioned fear response not only using aversive shocks, but also using threatening visual stimuli[39]; thus, pointing to a broad role of the vlPAG in fear learning.

Several electrophysiological and optogenetic studies indicate that vlPAG neurons encode positive fear prediction errors, determining the intensity of fear learning[40,42,46,47]. Since the optogenetic stimulation of the FN–vlPAG pathway specifically during CS–US pairing was sufficient to enhance the extinction of the fear response, indicating a decrease in the intensity of fear learning, this could be explained by an alteration of the positive prediction error signal during conditioning (Fig. 9). In our recordings, we found neurons in the vlPAG and FN that exhibited a strong reduction of firing during immobility and freezing, and an increase in the transfer of information between these FN and vlPAG neurons during, but also

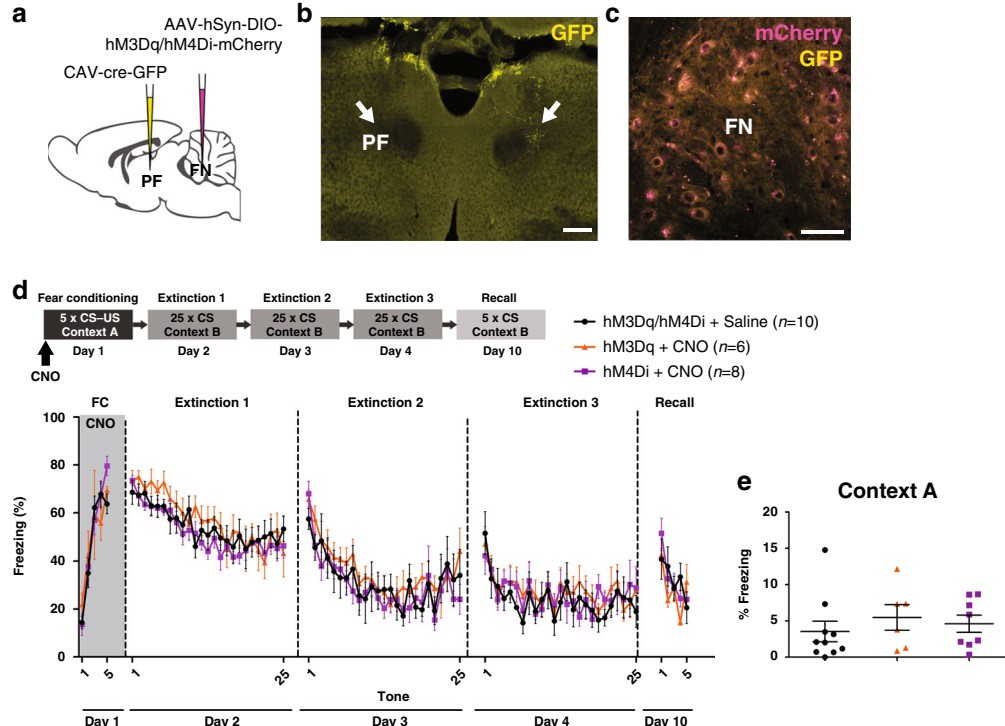

**Fig. 8 FN–PF pathway does not contribute to fear memory formation. a** Chemogenetic modulation of FN–PF-projecting neurons by bilateral cre-dependent expression of excitatory or inhibitory DREADDs (hM3Dq or hM4Di, respectively) in the FN in combination of CAV2-cre-GFP injection in the PF. **b** Injection site of CAV2-cre-GFP in PF (arrows, scale bar, 200 μm). **c** Cre-dependent expression of DREADDs in the FN (scale bar, 50 μm). **d** FN–PF pathway stimulation (hM3Dq + CNO, n = 6 mice) or inhibition (hM4Di + CNO, n = 8 mice) during fear conditioning had not significant effect on freezing behavior during the sessions (for stimulation: FC: $F(1,55) = 0.07$, $P = 0.795$; extinction 1: $F(1,323) = 3.03$, $P = 0.0826$, extinction 2: $F(1,267) = 3.87$, $P = 0.0503$, extinction 3: $F(1,279) = 3.65$, $P = 0.0571$; recall: $F(1,65) = 1.49$, $P = 0.226$; for inhibition: FC: $F(1,100) = 0.53$, $P = 0.468$; extinction 1: $F(1,405) = 3.86$, $P = 0.0502$, extinction 2: $F(1,247) = 0.76$, $P = 0.386$, extinction 3: $F(1,232) = 1.18$, $P = 0.279$; recall: $F(1,65) = 0.00$, $P = 0.956$; two-way ANOVA) compared to the control group (hM3Dq/hM4Di + saline, n = 10 mice). Data are presented as mean values ± SEM. **e** Percentage of freezing in context A before fear conditioning. Mice under inhibition (hM4Di + CNO, n = 8 mice) or activation (hM3Dq + CNO, n = 6 mice) of the FN-vlPAG pathway exhibit similar basal levels of freezing to the context A compared to the control group (hM3Dq/hM4Di + saline, n = 10 mice, df = 2, F = 0.435, P = 0.653, one-way ANOVA). Scatter dot plot with mean ± SEM. Source data are provided as Source data file.

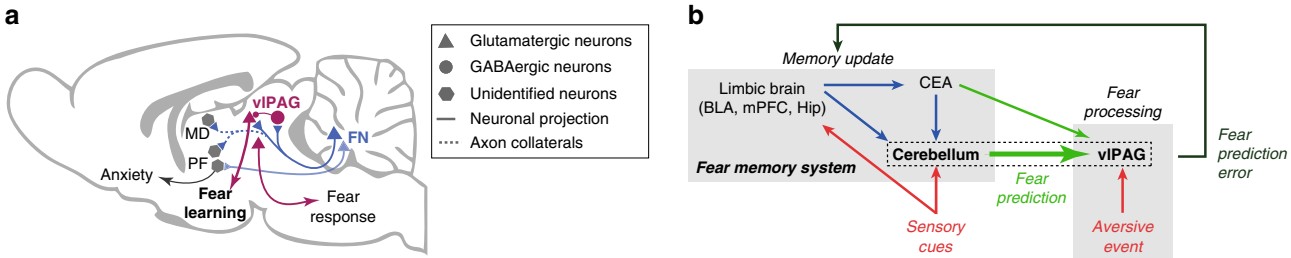

**Fig. 9 Summary of results and putative insertion of the FN–vlPAG pathway in fear learning process. a** Glutamatergic FN inputs form synaptic contacts on glutamatergic and GABAergic vlPAG neurons (see Figs. 1 and 2) that excite the vlPAG neurons (Fig. 3). FN–vlPAG axon collaterals (dotted lines) also target several thalamic nuclei involved in limbic circuits (PF parafascicular, MD mediodorsal; Supplementary Fig. 7). vlPAG-projecting FN neurons control fear learning rather than fear responses (Figs. 4 and 5, and Supplementary Figs. 4 and 5)—this is consistent with the observation that vlPAG does not contact freezing-inducing vlPAG neurons[26]. PF-projecting FN neurons rather control anxiety (Fig. 8 and Supplementary Fig. 8). **b** Schematics of the putative insertion of the cerebellum–vlPAG pathway studied here (dotted rectangle) in the fear prediction circuits: the cerebellum receives inputs from the sensory and emotional system[56]. The central amygdala, CEA, which carries fear predictions to the vlPAG[12], also projects to the cerebellum[25]. Departure from the predictions yields a fear prediction error[15], which drives the update of the fear memory. The cerebellum could thus participate to the propagation of fear predictions to the vlPAG: potentiated cerebellar output yields stronger prediction of US following CS, thus lower error at the occurrence of US, hence weaker learning of CS-US association during conditioning. Conversely, inhibited cerebellar output yields weaker prediction following CS, stronger error at the occurrence of US, and stronger CS-US association. N.B.: the aversive effect of US and the fear response to US are preserved by cerebellar manipulations.

after, conditioning. However, it remains unclear whether these neurons encode fear prediction error[41,42], notably since the CS and US usually induced motor responses which may also explain the increased firing. The elaboration of a fear prediction error signal in

the cerebellum shall recruit a large network, possibly involving the amygdala (Fig. 9). Indeed, the amygdala and the vlPAG have reciprocal connections that are essential for fear prediction error treatment[12,42]. However, the amygdala also projects heavily to the

ponto-cerebellar system and is essential for cerebellar CS activity in reflex conditioning[25], suggesting that a fear prediction signal could be provided by the amygdala both to the vlPAG and cerebellum. In addition, a recent imaging study has shown in a cued fear learning experiment that large portions of the cerebellar cortex are activated by fear prediction error[48]. Moreover, the participation of the cerebellum to shock prediction is illustrated in aversive eyeblink conditioning, where climbing fibers respond to the onset of the CS after learning[49]. Nevertheless, the nature of cerebellar computations in fear prediction is still unknown, but the cerebellum provides an extensive representation of the context thanks to the massive mossy fiber system and fine temporal learning, which may help tuning the fear prediction (Fig. 9).

Our study also indicates a modulatory role of the FN–vlPAG pathway in fear extinction. Optogenetic stimulation of FN–vlPAG inputs at the CS offset in extinction training was sufficient to enhance the extinction learning. Consistent with our results, evidence for negative fear prediction error, needed to drive extinction learning, was found in vlPAG[45], as well as in the ventral tegmental area and dorsal raphe[50,51], which also receive inputs from the vlPAG.

A striking finding of our work was that FN–vlPAG projections send collaterals to different thalamic nuclei, such as regions of the PF thalamus that projects to the BLA (Fig. 9). Nevertheless, FN–PF projections did not exert the modulation on fear learning found in FN–vlPAG neurons. In contrast, FN–PF activation increased anxiety in multiple anxiety tests, pointing toward an involvement of FN–PF in the modulation of anxiety states, which could be related to the reported contribution of PF to observational social fear[52].

Understanding how cerebellar circuits engage with processing and associative learning of emotions in other areas of the central nervous system is an important avenue for future research[17,25]. Indeed, the cerebellar involvement in reward processing is engaging a pathway between the cerebellum and the ventral tegmental area[17]. Our work shows that another pathway linking the cerebellum to vlPAG is controlling fear learning (Fig. 9). The study of this pathway could be particularly relevant for the deeper understanding of emotional control and maladaptive threat processing.

## Methods

**Animals**. Experimental subjects were adult C57BL/6 N male mice, 8–12-week-old, wild-type (Charles River Laboratories) or mutant male mice *Vglut2-cre* and *Glyt2-GFP* from an in-house colony (IBENS, Paris, France).

All mice were housed in groups of four mice per cage, room temperature controlled at 21–22 °C, humidity between 40 and 50%, in a 12 h light/dark cycle (light from 7:a.m. to 7:p.m.), and all the experiments were performed in awake freely moving mice during the light cycle. Food and water were available ad libitum. All animal procedures were performed in accordance with the recommendations contained in the European Community Council Directives (authorization number APAFIS#1334-2015070818367911 v3).

**Surgeries and stereotaxic injections**. Male mice 7–8-weeks old received a dose of buprenorphine and 15 min later they were deeply anesthetized with isoflurane 3%, and placed in a stereotaxic frame (Kopf Instruments). Anesthesia was kept constant with 1.5–2% isoflurane supplied via an anesthesia nosepiece, and body temperature maintained at 36 °C with a heating pad controlled by rectal thermometer. A local analgesia with 1 ml of 0.02% of lidocaine was injected subcutaneously above the skull, and then the skull was exposed and perforated with a stereotaxic drill at the desired coordinates relative to Bregma. For viral delivery, a capillary was lowered to coordinates just above the target area and 100–250 nl of virus solution was infused. After infusion, capillary was kept at the injection site for five more minutes before being slowly withdrawn. For viral injections and implants, the stereotaxic coordinates used were: FN: AP: −6.37, ML: ±0.70, DV: −2.12; vlPAG: AP: −4.45; ML: ± 0.55; DV: −2.50 to −2.40; PF: AP: −2.10; ML: ±0.70; DV: −3.42 to −3.00; BLA: AP: −1.35; ML: ±3.43; DV: −4.0; depths were taken relative to the dura.

**Neuroanatomical tracing**. For neuroanatomical tracing, mice were injected with 100 nl of anterograde AAV1-CB7-Cl-mCherry-WPRE-rBG (Upenn Vector) in the FN ($n = 14$), 100 nl CTB-alexa 488 or -alexa 555 (Invitrogen) in the BLA ($n = 6$),

and 250 nl CTB-CF594 (Biotium) or retrograde AAV-Syn-eGFP (Addgene) were injected in the vlPAG ($n = 6$). For synaptophysin labeling, 200 nl of AAV8.2-hEF1a-DIO-synaptophysin-GFP (Massachusetts General Hospital) was injected in FN of *vglut2-cre* mice ($n = 7$), and a group of these mice were injected with 300 nl of AAV1.CAG.Flex.tdTomato.WPRE.bGH (Upenn Vector) in vlPAG ($n = 4$), while the rest were used for gad67 immunostaining.

To study FN–vlPAG axon collaterals, mice were injected with 200 nl of retrograde AAV-cre-EBFP (Addgene) in the vlPAG, AAV1.CAG.Flex.tdTomato.WPRE.Bgh, and 200 nl of AAV8.2-hEF1a-DIO-synaptophysin-GFP in FN.

For the CTBs infusions, mice were perfused the fourth day after the injections. For the AAVs infusions, mice were perfused 21 days after the injections. Mice were anesthetized with ketamine 80 mg kg⁻¹ and xylazine 10 mg kg⁻¹, i.p., and the perfusions were performed with formalin (Sigma). Dissected brains were kept in formalin solution overnight at 4 °C, and the brains were then stored in PBS solution at 4 °C. Brains were sliced entirely at 90 μm using a vibratome (Leica VT 1000 S), and mounted on gelatin-coated slides, dried, and then coverslipped with Mowiol (Sigma). For counterstaining, sections were coverslipped with a mounting medium with DAPI (Fluoroshield, Sigma). Slices were analyzed and imaged using a confocal microscope (Leica TCS Sp8), and images were edited and analyzed using Image J. In the same way, the placement of the optic fiber, electrodes, and injections sites for all the experiments were assessed when the experiment was completed. Mice with no viral expression, or misplacement of the optic fiber or electrode were excluded from the analysis.

**Immunohistochemistry**. For the identification of GABAergic neurons in the vlPAG, mice were perfused with PFA 4%, stored overnight in PFA 4% at 4 °C, and then stored in 25% sucrose solution in PBS. Brains were sliced coronally at 50 μm using a vibratome and slices were stored at −20 °C in a cryoprotective solution until the moment of the immunostaining.

In order to analyze neuronal activity, immunostaining of the immediate early gene c-Fos was performed. Mice underwent fear conditioning protocol and 50 min after the end of the protocol each mice was perfused as has been described above.

For the immunostaining, selected slices were washed with PBS-0.3% Triton X-100, blockage of nonspecific sites was assessed by 2 h of incubation with 3% normal donkey serum (NDS) for GAD67-immunostaining ($n = 3$) or with normal goat serum (NGS) for c-Fos-immunostaining ($n = 10$). Sections were then incubated in a solution containing mouse anti-GAD67 (1:500, Milipore #MAB5406) in PBS-0.3% Triton X-100 with 1.5% NDS at 4 °C for 72 h; or rabbit polyclonal anti-c-fos (1:800, Milipore #ABE457) in PBS-0.3% Triton X-100 with 1.5% NGS at 4 °C for 24 h. After first antibody incubation, slices were rinsed and sections were 2 h incubated at room temperature with secondary antibody, donkey anti-mouse IgG conjugated to Alexa Fluor 488 or to Alexa 555 (1:400, Invitrogen), or goat anti-rabbit IgG-FITC (1:300, Jackson Immunoresearch). Slices were mounted on gelatin-coated slides with Mowiol and analyzed with the confocal microscope.

**Chemogenetics**. To specifically modulate the activity of FN neurons projecting to vlPAG, excitatory or inhibitory DREADDs were expressed by bilaterally injection (200 nl per injection site) of pAAV-hSyn-DIO-hM3Dq-mCherry (cre-dependent expression of excitatory DREADD, Addgene) or pAAV-hSyn-DIO-hM4Di-mCherry (cre-dependent expression of inhibitory DREADD, Addgene) in the FN in combination of 300 nl CAV2-cre-GFP (Plateforme de Vectorologie de Montpellier) viral infusion in vlPAG. Surgeries and injections were performed 2 weeks before mice performed the behavioral tests. For neuronal modulation of animals expressing DREADDs, Clozapine N-oxide (Tocris Bioscience, 1.25 mg kg⁻¹) was diluted in saline and injected i.p. 30 min before the start of the experimental session. Control group was injected with saline. Animals that received this treatment in experiments were habituated by saline injections during handling sessions.

Another group of mice called sham, underwent the same surgery procedures included the insertion of the cannula without withdrawal of virus. To evaluate whether CNO had an effect in absence of DREADD expression, these mice were injected with saline or CNO before performing the behavioral tests.

**Pavlovian fear conditioning and extinction**. Pavlovian fear conditioning and extinction took place in two different contexts (A and B), located inside of a sound attenuating box (Ugo Basile). Pavlovian fear conditioning was conducted in a context A: 17 × 17 × 25 cm square chamber with black/white-checkered walls and a grid floor, with peppermint-soup odor. The chamber was cleaned with a 70% ethanol within mice. Following a 180 s acclimation period, mice received five pairings between a 30 s, 80 dB, 2.7 kHz tone (CS) and a 0.5 s, 0.4 mA electrical footshock (US), in which the US was presented during the last 0.5 s of the CS, and with an ITI of 120 s after each CS–US pairing. Stimuli presentation were controlled by the Ethovision XT 14 (Noldus).

Twenty-four hours later, recall of the response to the CS and the first extinction training were performed. Mice were placed in a novel context B: cylindrical chamber with yellow semitransparent wall, an opaque solid-Plexiglas floor, and vanilla-extract solution to provide a distinctive olfactory cue. The chamber was cleaned with the disinfectant detergent solution Surfa'safe premium (Anios) between different run. Following an initial 180 s acclimation period, the mouse received 25 × 30 s presentations of the CS alone (30 s no-stimulus interval). The

second and third extinction sessions took place in the next 2 days under the same conditions. Finally, on day 10, recall test was performed in context B, by $5 \times 30$ s presentations of the CS (30 s no-stimulus interval).

Mice were videotracked using Ethovision XT 14 during all the trial. Freezing (no visible movement except that required for respiration) was assessed by the inactive periods, defined as periods of time during which the average pixel change of the entire video image was <0.5 % (from one video frame to another one), and the threshold was fixed avoiding the detection of the breathing movements. Freezing behavior was analyzed during each CS presentation, and during the habituation period to the context A and B.

**Anxiety tests**. *Open field test.* To evaluate anxiety-like behavior and locomotor activity, mice were placed in a circular arena (38 cm diameter, Noldus) and videotracked from above. The position of the center point of the mice was tracked with Ethovision XT 14 and the distance moved, time in the center area, frequency of entries in the center area, were analyzed for a 20 min period.

*Elevated plus maze.* Mice were placed in the center zone ($6 \times 6$ cm), facing an open arm of an elevated plus maze (Noldus, elevated 52 cm above the floor) with two open arms (36 cm length, 6 cm width) and two wall-enclosed arms (closed arms, 36 cm length, 6 cm width, walls 25 cm high), and let explore freely for 5 min. Their path was videotracked using Ethovision XT 14, and the amount of time spent and distance moved in the open arms, closed arms, and center zone were analyzed.

*Light–dark box.* Mice were placed in the light zone and let to explore the light–dark arena (Noldus) freely for 5 min. Light zone was $40 \text{ cm} \times 20 \text{ cm}$ in size, 150 lux, while dark zone was $20 \times 20$ cm and ~0 lux. Their path was videotracked in the light and dark zones using Ethovision XT 14, the amount of time spent in the light versus dark zones and distance traveled in the light zone were evaluated, as well as the latency until they escaped to the dark zone.

**Hot plate and tail-immersion tests**. Effect on sensitivity was tested 1 week after recall test using a hot plate analgesia meter (Harvard apparatus) and tail-immersion test. Mice received a CNO or saline i.p. injection and 30 min after were put on the hot plate at 55 °C. The experiment was stopped as soon as the mice performed the first licking or jump. Mice were videotracked and the latency to the first reaction (hindpaw shaking or licking, or jump) was recorded.

Tail-immersion test was performed after hot plate test. Mice tale tips were immersed in hot water with a temperature of 50 °C, and tail withdrawal latency was recorded.

**Optogenetic stimulation**. For optical manipulation, mice were injected bilaterally with 200 nl of cre-dependent AAV-ChR2-GFP (Addgene) in the FN combined with local injection of 200 nl of retrograde AAV-cre-EBFP into the vlPAG. To perform optical stimulation, mice were implanted bilaterally with an optic fiber (0.22 numerical aperture, Thor labs) housed in a stainless steel ferula (Thor Labs) just dorsal to the FN. Ferrules were adhered to the skull with adhesive, and then a protective head cap was constructed using dental cement. The viral injections and optic fiber implantations were performed in the same surgery, and the optical manipulation and fear conditioning were performed 3 weeks after surgery to ensure ChR2 expression. At the end of the experiments, optic fiber positions were verified.

For the optogenetic activation of the FN during fear conditioning or extinction, mice were randomly designated to non-stimulated or stimulated groups. The stimulated groups received 20 ms blue light pulses, at 10 Hz, 1 mW during a total of 6 s. The CS–US group received the light stimulation on each of the five CS–US presentations, from the last 3 s of the CS until 3 s posterior to the end of CS–US presentation; the ITI group received the light stimulation randomly within the ITI, 30 s after and before the CS, during fear conditioning. For CS offset group, mice received light stimulation from the last 3 s of the CS until 3 s posterior to the end of CS presentation during extinction sessions 1 and 2.

**Optogenetics and electrophysiological recordings**. For optical manipulation combined with electrophysiological recordings, electrodes were implanted in mice injected bilaterally with 200 nl of cre-dependent AAV-ChR2-GFP in the FN, and with 200 nl of retrograde AAV-cre-EBFP into the vlPAG, and implanted uni-laterally with an optic fiber in the FN (see "Optogenetic stimulation" section). The cerebellar FN preferentially sends projections to the contra-lateral vlPAG regions (Fig. 1a, b). Therefore, in order to compare responses, we recorded neuronal activity in both contra- and ipsi-lateral vlPAGs when we unilaterally stimulated the FN. To record cell activity in the vlPAG, we used bundle of electrodes consisting of nickel/Chrome wire (0.004 inches diameter, Coating 1/4 Hard PAC) folded and twisted into bundles of six electrodes. Pairs of bundles were inserted in metal cannulas (stainless steel, 29 G, 8 mm length, Phymep), in order to protect them and to guarantee a correct placement into the brain. Cannulas and bundles were also attached to an electrode interface board (EIB-16; Neuralynx, Bozeman) by Loctite universal glue in a configuration allowing us to record vlPAG. The microwires of each bundle were connected to the EIB with gold pins (Neuralynx). The entire EIB and its connections were fixed in place by dental cement (Super Bond). The impedance of each electrode was measured and the 1 kHz impedance was set to 200–500 kΩ, using gold-plating (cyanure-free gold solution, Sifco). During the

surgery, the electrode bundles were inserted into the brain and the ground was placed over the cerebellum, between the dura and the skull. The viral injections, optical fiber and electrodes implantation were performed in the same surgery, and the optical manipulation and recordings were performed 3 weeks after surgery to ensure ChR2 expression.

For the ramp of optical activations, electrophysiological recordings in the vlPAG were performed in freely moving mice placed in an open field. Recordings were performed using an acquisition system with 16 channels (sampling rate 25 kHz; Tucker-Davis Technology System 3, Tucker-Davis Technologies) as described[53,54]. Mice received a stimulation at 0.5 Hz, 50 ms, and the light intensities used were 0.37, 0.54, 0.75, 1.03, and 1.33 mW (at the fiber tip). The signal was acquired by a headstage and amplifier from TDT (RZ2, RV2, Tucker-Davis Technologies). The spike sorting was performed with Matlab (Mathworks) scripts based on $K$-means clustering on PCA of the spike waveforms (Paz et al. 2006). At the end of the experiments, the position of the electrodes was verified by electrolytic lesions and comparison to a brain atlas after brain slicing was performed.

**Chemogenetics and electrophysiological recordings**. Mice were injected bilaterally with 300 nl CAV2-cre-GFP in the vlPAG, and with 200 nl of pAAV-hSyn-DIO-hM3Dq-mCherry or pAAV-hSyn-DIO-hM4Di-mCherry in the FN, as has been detailed above (see "Chemogenetics" section). After viral infusions, electrodes were placed in FN and vlPAG as was described above (see "Optogenetics and electrophysiological recordings" section). Fear conditioning was performed 2–3 weeks after the surgeries, and electrophysiological and inertial recordings during fear conditioning were analyzed. Gyroscopic data were acquired using the MultiChannelSystems wireless headstage W2100-HS16 embedded gyroscope. The motor activity signal was computed as the norm of the rotation vector, smoothed by convoluting it with a Gaussian kernel (s.d. = 64 ms). The smoothed signal was then binarized by applying a threshold of 12 degree $\text{s}^{-1}$ between active and immobile states.

**Statistical analysis**. The effect of activation or inhibition of FN–vlPAG projections was analyzed using analysis of variance (ANOVA) and Newman–Keuls or Bonferroni post hoc comparisons, or paired $t$ test, where appropriate. All statistics were performed in Graph Pad Prism® (Version 5) and RStudio® (R version 3.6.3 and RStudio version 1.2.5), unless otherwise indicated, and all statistical tests used are indicated in the figures legends. The effect of trial on freezing during fear conditioning and extinction was analyzed using two-way ANOVA. Experimental designs with two categorical independent variables were assumed to be normal and analyzed by two-way ANOVA without formally testing normality. All significance levels are given as two-sided and were corrected for multiple comparisons, wherever applicable. Statistical significance was set at $P < 0.05$.

To analyze the response to the optogenetic stimulations, we constructed the PSTH (bin 2 ms) and converted these PSTH to a $z$-score using the pre-PSTH period (subtracting by baseline mean and dividing by the baseline standard deviation), the latency was determined by the time of the first bin with strongly significant $z$-score (>4). Classification of cells in two sets based on the link between firing and immobility was performed by computing the PSTH (bin 100 ms) around the onset of immobility episodes, irrespective of their duration. The PSTH were normalized by their average and clustered in two groups using $K$-mean algorithm. The average firing rate during immobility and active behavior was simply computing by dividing the number of spikes in each of these states by the total duration spent in each of these states.

Information theory methods were used to examine the relation between cell firing and binarized motor activity. Spiketrains of neuronal activities were binned using bins of 10 ms. To obtain a matching motor signal, the binarized motor activity was interpolated and resampled at every time point corresponding to the bin timestamp (left edge of the bin). To study directional influences, we related the source signal to the target signal one time bin (10 ms) later. The transfer entropy[43] was calculated using the PyIF Python package (v0.1), allowing the computation of bivariate transfer entropy between two binned spiketrains, or between a binned spiketrain and the corresponding motor signal. The PID was used to decompose the influence of the neuronal activity in FN and locomotion on the neuronal activity in vlPAG. The PID was computed using the iDTxl Python package (v1.0)[55], as a decomposition of the multivariate mutual information at lag −10 ms between the two sources signals and the target signal, using the Goettingen estimator.

**Statistics and reproducibility**. For anatomical tracing in Fig. 1b, d–f, analysis was repeated independently in eight mice showing similar results. For Fig. 2b–d, f–h, independent analysis of three mice exhibited similar results. For fear conditioning and extinction experiments in DREADD mice, each experiment was repeated independently at least four times, and pooled together (Figs. 4d, e, 6b and 8, and Supplementary Figs. 3 and 8). Viral injection site and expression were checked in each mouse included in the results, showing similar results (Figs. 4b, c, 5b, c and 8b, c). For electrophysiological recordings, experiments were repeated independently two times, and viral expression and electrode positions were checked showing similar results (Fig. 5b, c). For optogenetics (Figs. 4g, i and 6e, and

Supplementary Fig. 5), each experiment was repeated independently two times, obtaining similar results; thus, the results were pooled together. Viral expression and optic fiber position were checked in each mouse included in the results, showing similar results (Fig. 4g). For anatomical tracing in Fig. 7b, c, e, analysis was repeated independently in four mice showing similar results. For anatomical tracing in Supplementary Fig. 1b–e, the experiment was carried out three independently times, for Supplementary Fig. 1g, two independently times, and for Supplementary Fig. 7b-f was repeated four independently times, all showing similar results.

**Reporting summary**. Further information on experimental design is available in the Nature Research Reporting Summary linked to this paper.

## Data availability
The data that support the findings of this study (Figs. 1–9, Supplementary Figs. 1–8) are available from the corresponding author upon reasonable request. For additional information, please refer to the corresponding Life Sciences Reporting Summary. Source data are provided with this paper.

## Code availability
The code for analysis is available from the corresponding author upon reasonable request.

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

## Acknowledgements

This work was supported by Fondation pour la Recherche Medicale (FRM, DPP20151033983) to D.P., and Agence Nationale de Recherche to D.P. (ANR-16-CE37-0003-02 Amedyst, ANR-19-CE37-0007-01 Multimod, Labex Memolife) and to C.L. (ANR-17-CE37-0009 Mopla, ANR-17-CE16-0019 Synpredict) and by the Institut National de la Santé et de la Recherche Médicale (France). I.A.G. was supported by the project 819/11.01.2019 by UMF Carol Davila, Bucharest. We thank the Imaging Facility at IBENS (IMACHEM-IBiSA, France-BioImaging ANR-10-INBS-04, FRC Rotary International France, Investments for the future, ANR-10-LABX-54 MEMOLIFE).

## Author contributions

J.L.F. and D.P. designed the experiments; J.L.F., C.L., and D.P. wrote the manuscript; J.L.F., R.W.S., and C.L. analyzed the data; and J.L.F., H.B.A., I.A.G., R.W.S., and C.M.H. performed the experiments.

## Competing interests

The authors declare no competing interests.
