## [Peer Review File · Nature Communications]

Reviewers' comments:

Reviewer #2 (Remarks to the Author):

Using a variety of anatomical, electrophysiological, and perturbation methods, the main claims of the manuscript are that first that vglut2 cerebellar FN neurons project to both excitatory and inhibitory neurons vIPAG and that this (but not a secondary pathway through the thalamus) can modulate fear learning and extinction through a mechanism of increased electrophysiological coupling (demonstrated through cross correlation analysis). Second, they show using pathway specific DREADD Gi/Gq manipulations and some pathway specific optogenetics, that fear learning and extinction can be disrupted. Using in vivo recordings in both the FN and vIPAG, they show that the response properties to the CS during and extinction paradigm are heterogenous in both areas, with some neurons showing short onset responses, and others showing sustained or inhibited responses. Together, this paper begins to piece together a novel cerebellar-midbrain circuit encoding the representation of the CS during fear learning and extinction that can bidirectionally control the expression of freezing behavior by modulating the "strength" of the fear memory. While the author's use of pathway specific methods viral methods to elucidate the contribution of a single pathway is to be applauded, other aspects of this paper require more thought, and the overall logic of the paper is difficult to follow. The authors wish to frame the data in the context of prediction error, but the response to the US is never shown and it is unclear which of the heterogeneous types of the of neurons recorded in the FN actually projects to the vIPAG.

Major concerns:

Electrophysiology:

While the recordings performed in two brain regions are a strength of this paper, the choice to evaluate the responses only aligned to the CS is puzzling, since vIPAG neurons are also drive and are responsive to freezing. No effort was made to disentangle the sensory response across before and after learning, or to test whether cerebellar inputs have are responsive to both sensory and behavior variables. Understanding whether both FN and vIPAG neurons are driven by US or behavior responses is critical to understanding how this circuit might drive learning.

In addition, while the authors show that the downstream vIPAG neurons are activated by FN Chr2 stimulation, a more relevant experiments would have been to use the same method to optotag vIPAG projecting neurons in the FN, which would reveal the precise nature of the input.

While the cross-correlation experiments in Figure 6 are potentially interesting, more analysis is required to detect whether correlation increases are more than spurious. Correlations should be presented separately for evoked and spontaneous activity (after accounting for differences in freezing behavior or other fear behaviors), and correlations should be compared to a control correlation with the same number of spikes jittered in time.

Quantification: In many sections of this paper, critical controls are missing and inappropriate statistical comparisons are being made.

For example, in most DREADD experiments, CNO modulated cohorts should be compared not just with a saline cohort (with the same viral construct), but with a CNO only control. This appears to be done in a spotty way across the paper and is not consistent across the paper.

Quantification of the anatomical claim that FN neurons synapse onto both excitatory and inhibitory neurons in the vIPAG in figure 2 is completely absent.

Optogenetics and DREADD:

The use of DREADDs rather than optogenetic manipulations in this study is unfortunate because it

precludes a more systematic inquiry of how the vIPAG is using the FN signal. While it appears the direction of the result from the DREADD and the optogenetics experiments goes the same direction (increased activity reduces freezing), it is unclear why the optogenetic manipulation is so much weaker with later onset than the DREADD activation. In addition the optogenetic manipulation would be best compared to stimulation delivered at other timepoint (for example during freezing, or at random time points), in order to evaluate whether the CS-US juncture is the critical timepoint.

In addition, since the data seems to suggest that the incidence "group 1" may increase with extinction, it would be good to test whether optogenetically stimulating during the CS during extinction would also reduce freezing.

All effects in Supp fig 5 also missing the critical CNO control, and are not corrected for multiple comparisons.

Collaterals:

Though the authors show other pathways that target the vIPAG, the more critical experiment is to test whether the vIPAG projecting FN neurons send collaterals to other pathways. For example, do neurons that project to the vIPAG also project to the thalamus? This would allow us to evaluate whether pathway specific manipulations are truly pathway specific.

The order of the figures makes the logic of the paper difficult to follow, since electrophysiology is split into separate parts of the paper, and the data from the cerebellar to thalamic pathway is confusingly sprinkled throughout.

Minor:

Writing: typo in Supp Fig 5. Should be FN-PF

Ipsi vs Contra fos for supp figure 4 should be separately quantified

Reviewer #3 (Remarks to the Author):

In this manuscript the authors evaluated the contribution of a cerebellar fastigial nucleus (FN) projection onto ventrolateral periaqueductal grey (vIPAG) neurons during fear acquisition and extinction. Using a combination of state of the art approaches, the authors first demonstrated that roughly 12% of FN excitatory neurons directly and monosynaptically project to vIPAG excitatory and inhibitory neurons. Next, using chemogenetic approaches they revealed that the modulation of FN inputs can bidirectionally control fear and extinction learning. In parallel they demonstrate that FN inputs onto thalamic neurons projecting to the basolateral amygdala also play a role in the control of fear behavior through the modulation of fear memory expression. These effects were not dependent on anxiety levels or pain sensitivity. The authors concluded that the cerebellum, through an excitatory FN-vIPAG projection, potentially controls fear memory.

The concept proposed in this manuscript is very appealing as it challenges the current dogma on the organization of fear circuitry centered around the amygdala. Although earlier studies have suggested a role of the cerebellum in the control of fear and extinction behavior, this study goes a step further and delineate a direct cerebellar-brainstem circuit controlling fear and extinction acquisition independently of the amygdala. I reviewed a first draft of this manuscript a while ago and the current version has been dramatically improved and all of my concerns have been addressed.

I have only a remaining minor comment to the authors

The effect of the manipulation of the FN-PF pathway is small and I am not really convinced this is a

true effect on fear expression as compared to interindividual variability. I will tone down a bit this statement unless the authors show a fear expression effect on a subsequent retrieval test.

Responses to reviewers

We are extremely grateful to the reviewers for their carefully reading and for their very constructive comments. We did our best to improve the manuscript under their guidance and we feel it has immensely benefited from their remarks and requests. The manuscript has been substantially revised in all its sections, we have added new experiments and analysis, and introduced new figures, figure panels and supplementary figures.

We added new groups to assess the impact of FN-vIPAG optogenetic stimulations at the offset of the CS during extinction and between CS during fear conditioning. We also performed experiments on a new batch of mice with FN-vIPAG optogenetic stimulation at the time of CS-US pairing to increase the number of each group.

We succeeded in performing telemetric (to avoid electrical damage to the recording system) simultaneous electrophysiological recordings in the FN and vIPAG on the day of conditioning. This revealed intricate motor and sensory responses, in which we found increased FN-vIPAG communication during the conditioning stage.

We quantified the GABAergic and the glutamatergic neurons in vIPAG contacted by synapses boutons from FN and confirmed the high rate of connections in regions of the vIPAG.

We searched and, although these were scarce, we found some axon collaterals of FN-vIPAG projecting neurons in the thalamus; testing each of the regions with collaterals is beyond reach, but we focused on the parafascicular thalamus which may likely relay FN projections to the amygdala, a key player in fear learning. The chemogenetic manipulations of the FN-PF neurons did not affect fear learning indicating that the collaterals of FN-vIPAG neurons to the PF exert a limited effect on fear learning. Instead, these experiments revealed an influence of FN-PF neurons on emotions (anxiety), which was distinct from the influence of FN-vIPAG neurons.

We also added new CNO control experiment in sham mice for extinction and anxiety tests, which confirmed the specificity of the effects reported.

Overall we feel our previous results are reinforced by the complementary experiments and analysis.

Please find below a point by point response to the reviewers comments.

Reviewer #2 (Remarks to the Author):

Using a variety of anatomical, electrophysiological, and perturbation methods, the main claims of the manuscript are that first that vglut2 cerebellar FN neurons project to both excitatory and inhibitory neurons vIPAG and that this (but not a secondary pathway through the thalamus) can modulate fear learning and extinction through a mechanism of increased electrophysiological

coupling (demonstrated through cross correlation analysis). Second, they show using pathway specific DREADD Gi/Gq manipulations and some pathway specific optogenetics, that fear learning and extinction can be disrupted. Using in vivo recordings in both the FN and vIPAG, they show that the response properties to the CS during and extinction paradigm are heterogenous in both areas, with some neurons showing short onset responses, and others showing sustained or inhibited responses. Together, this paper begins to piece together a novel cerebellar-midbrain circuit encoding the representation of the CS during fear learning and extinction that can bidirectionally control the expression of freezing behavior by modulating the strength of the fear memory.

While the authors use of pathway specific methods viral methods to elucidate the contribution of a single pathway is to be applauded, other aspects of this paper require more thought, and the overall logic of the paper is difficult to follow.

We edited the text to clarify the presentation. Currently, the paper is organized as follow:

- Histological demonstration of FN inputs to the vIPAG and identification of the targets in the vIPAG (Fig 1,2)
- Physiological observation of vIPAG activation following FN optogenetic stimulations in vivo to confirm functional connections. (Fig 3)
- Chemogenetic and optogenetic manipulations in conditioning modulates the strength of memories; the effect seem to take place during rather than after conditioning (Fig 4)
- Electrophysiological recordings during conditioning show that a fraction of FN and vIPAG neurons modulated by the motor state are also showing increased interaction during conditioning (Fig 5).
- Chemogenetic and optogenetic manipulations during extinction also indicate a role in extinction (Fig 6).
- The region receiving collaterals from the FN-vIPAG and projecting to amygdala (thus possibly mediating the effect on fear conditioning) is involved in anxiety but not in fear learning (Fig 7, 8).

The authors wish to frame the data in the context of prediction error, but the response to the US is never shown and it is unclear which of the heterogeneous types of the of neurons recorded in the FN actually projects to the vIPAG.

Major concerns:

Electrophysiology:

While the recordings performed in two brain regions are a strength of this paper, the choice to evaluate the responses only aligned to the CS is puzzling, since vIPAG neurons are also drive and are responsive to freezing. No effort was made to disentangle the sensory response across before and after learning, or to test whether cerebellar inputs have are responsive to both sensory and behavior variables. Understanding whether both FN and vIPAG neurons are driven by US or behavior responses is critical to understanding how this circuit might drive learning.

- We are very grateful for this comment. We managed to record the FN and vIPAG during conditioning (despite the high-voltage electric shocks of the US) and we accompanied these recordings with inertial measurements which allowed us to have an excellent temporal resolution on the mice movements/immobility. This helped to identify two populations of cells with different sensitivity to the movement. The cells which showed

responses to the CS and US where indeed the ones showing a sensitivity to motor activity. We also added directional analysis of the interactions between motor activity, FN and vIPAG firing before during after conditioning.

In addition, while the authors show that the downstream vIPAG neurons are activated by FN ChR2 stimulation, a more relevant experiments would have been to use the same method to optotag vIPAG projecting neurons in the FN, which would reveal the precise nature of the input.

- Opto-identification of projecting neurons is a difficult experiment in this system if requiring to have a good alignment of electrode positioning in the FN, injection sites in the vIPAG and FN (we nonetheless started these experiments which were stopped by the Covid crisis and are difficult to restart in our current conditions of limited operation).

While the cross-correlation experiments in Figure 6 are potentially interesting, more analysis is required to detect whether correlation increases are more than spurious. Correlations should be presented separately for evoked and spontaneous activity (after accounting for differences in freezing behavior or other fear behaviors), and correlations should be compared to a control correlation with the same number of spikes jittered in time.

- We performed the proposed controls on the correlations which are not likely spurious, but the observation of the modulation of firing by the motor state in the conditioning day (see above) strongly complicated the interpretation. Since we do not have well synced, high resolution measure of immobility during these extinction day recording (and since extinction is anyway a secondary topic of this manuscript) we removed these data from the new version of the MS.

Quantification: In many sections of this paper, critical controls are missing and inappropriate statistical comparisons are being made.

For example, in most DREADD experiments, CNO modulated cohorts should be compared not just with a saline cohort (with the same viral construct), but with a CNO only control. This appears to be done is a spotty way across the paper and is not consistent across the paper.

- We thank the reviewer for this comment; we performed more control experiments and we present now in a more consistent way the controls across the paper.
- In DREADD experiments, control group correspond to a pool of mice expressing one of the two constructs for DREADD (hM4Di or hM3Dq) and exposed to saline instead of CNO. As we showed in the Supplementary figure 3 a, where the saline control group were split in two according to their construct for DREADD, there were similar levels of freezing along all the sessions. This demonstrates that the expression of one or the other receptor has not effect *per se* in absence of its specific ligand CNO. Thus, we decided to pool them together in one saline group; this also sensibly simplifies the graph in the Figure 4.
- The CNO control group correspond to the one we called 'sham' group. The control group for CNO during fear conditioning had been included in the first version of the manuscript, and now we have added another CNO control group for extinction in Supplementary figure 3 b, and they were compared with the sham mice exposed to saline. There were not significant differences between these groups indicating that CNO

has not effect on behavior or fear memories in absence of DREADDs. Thus, we only keep the Control group injected with saline for comparisons.

- We have also added the CNO control on sham mice for the anxiety tests in Supplementary figure 8c.

Quantification of the anatomical claim that FN neurons synapse onto both excitatory and inhibitory neurons in the vIPAG in figure 2 is completely absent.

- As the reviewer asked, quantification of vGluT2 and GAD67 neurons exhibiting appositions of synaptophysin FN terminal boutons have been made and reveal high connection rates in the fields with dense FN inputs. The graphs have been added to the Figure 2.

Optogenetics and DREADD:

The use of DREADDs rather than optogenetic manipulations in this study is unfortunate because it precludes a more systematic inquiry of how the vIPAG is using the FN signal. While it appears the direction of the result from the DREADD and the optogenetics experiments goes the same direction (increased activity reduces freezing), it is unclear why the optogenetic manipulation is so much weaker with later onset than the DREADD activation. In addition the optogenetic manipulation would be best compared to stimulation delivered at other timepoint (for example during freezing, or at random time points), in order to evaluate whether the CS-US juncture is the critical timepoint.

- The reviewer refers to the magnitude of the results by two very different approaches, which involve different viral constructs and different expression efficacy. Moreover, the optogenetic stimulation is made through an optic fiber that very likely will not target all the ChR2-expressing neurons in the FN (which is much larger than the 200 μ m diameter optical fiber), contrarily to the chemogenetic experiment in which CNO is administrated systemically and it may reach all neurons expressing DREADDs. Considering all this, it would be expected to have different intensities on the effect in the behavior between these two approaches. Nevertheless, we have repeated the optogenetic experiment in order to increase the sample and we were able to show more clearly the differences compared to its control.
- We have also analyzed the effect of the optogenetic stimulation randomly within the ITI (between tones and after US), and we have found that it has an effect but in this case on the curve of fear learning (significantly lower) indicating a lower fear during conditioning. Consistent with this, we found lower levels of freezing during the retrieval in extinction 1 (Supplementary figure 5). Nevertheless, this effect would not be the same than the one obtained during the CS-US pairing which does not affect the freezing during learning.
- Overall we believe optogenetic and DREADD experiments are complementary, the former being more resolved in time (but possibly creating over-synchronous patterns

with little physiological relevance) and the DREADD providing a prolonged shift in excitability which more likely respects the physiological patterns of firing compared to optogenetic experiments. In the current stage, we feel that the convergence of the results obtained during fear conditioning by the two approaches reinforces the findings.

In addition, since the data seems to suggest that the incidence group 1 may increase with extinction, it would be good to test whether optogenetically stimulating during the CS during extinction would also reduce freezing.

- We thank the reviewer for this suggestion. The suggested experiment indeed revealed the effect of optogenetic stimulation during the CS offset during extinction learning and we have found a robust effect on the extinction learning, so we have added this in the Figure 6c-e. Contrarily to the fear learning, the sign of the effects on fear extinction of excitatory DREADD and of the optogenetic stimulation are different. This suggests that the effect on extinction strongly depends on the temporal pattern of activation of the FN-vIPAG neurons.
- We think that these experiments are opening another field of investigation, since fear extinction involves more brain regions and imply different mechanisms than fear learning. Indeed publications in the field often investigate either learning of extinction. We feel that, with the reviewers permission, the results of the figure 6 could be removed from the MS since they are secondary to the main effect studied (fear learning).

All effects in Supp fig 5 also missing the critical CNO control, and are not corrected for multiple comparisons.

- This electrophysiology figure has been removed. This comment raises the question of the different control group. Since the sham control CNO group has not indicated any effect neither on freezing or anxiety behavior, nor on the fear and extinction learnings, nor on memory formation and retrieval, we did not add SALINE and CNO groups of Sham mice without DREADD injections for electrophysiological recordings.

Collaterals:

Though the authors show other pathways that target the vIPAG, the more critical experiment is to test whether the vIPAG projecting FN neurons send collaterals to other pathways. For example, do neurons that project to the vIPAG also project to the thalamus? This would allow us to evaluate whether pathway specific manipulations are truly pathway specific.

- We have added the analysis of axon collaterals for FN-vIPAG projecting neurons in the thalamus in Supplementary figure 7. Indeed, we found that FN neurons projecting to vIPAG have collaterals to other areas in the thalamus. Testing the extent of the contribution of collaterals is currently difficult in the current state of technology; light stimulation of the terminals in the target regions may produce antidromic spikes which will invade the collaterals.
- The correspondence between the area of PF containing the soma of neurons projecting to the amygdala and the area receiving some collaterals from the FN-vIPAG neurons

raises the concern that the FN-vIPAG neurons could exert their effect on learning via these collaterals. Yet, our study of FN-PF projections failed to reveal an effect on fear learning, suggesting a minor –if any- contribution of these collaterals to fear learning. Instead, this revealed another contribution of the FN to the emotional system by a contribution to anxiety via a FN-PF pathway.

The order of the figures makes the logic of the paper difficult to follow, since electrophysiology is split into separate parts of the paper, and the data from the cerebellar to thalamic pathway is confusingly sprinkled throughout.

- We have changed and rearranged the figures, as well as their order in the text, in order to add the new results and to improve the understanding of the paper (see summary above).

Minor:

Writing: typo in Supp Fig 5. Should be FN-PF

- Corrected.

Ipsi vs Contra fos for supp figure 4 should be separately quantified

- For all chemogenetic experiments, mice were injected with DREADDs bilaterally in the FN, thus the effects of stimulation and inhibition of FN-vIPAG projections were bilaterally also at the vIPAG. In order to clarify this point, we have included ‘bilateral FN-vIPAG chemogenetic stimulation and inhibition’ in the figure legend (Supplementary figure 4).

Reviewer #3 (Remarks to the Author):

In this manuscript the authors evaluated the contribution of a cerebellar fastigial nucleus (FN) projection onto ventrolateral periaqueductal grey (vIPAG) neurons during fear acquisition and extinction. Using a combination of state of the art approaches, the authors first demonstrated that roughly 12% of FN excitatory neurons directly and monosynaptically project to vIPAG excitatory and inhibitory neurons.

Next, using chemogenetic approaches they revealed that the modulation of FN inputs can bidirectionally control fear and extinction learning. In parallel they demonstrate that FN inputs onto thalamic neurons projecting to the basolateral amygdala also play a role in the control of fear behavior through the modulation of fear memory expression. These effects were not dependent on anxiety levels or pain sensitivity. The authors concluded that the cerebellum, through an excitatory FN-vIPAG projection, potentially controls fear memory.

The concept proposed in this manuscript is very appealing as it challenges the current dogma on the organization of fear circuitry centered around the amygdala.

Although earlier studies have suggested a role of the cerebellum in the control of fear and extinction behavior, this study goes a step further and delineate a direct cerebellar-brainstem

circuit controlling fear and extinction acquisition independently of the amygdala. I reviewed a first draft of this manuscript a while ago and the current version has been dramatically improved and all of my concerns have been addressed.

I have only a remaining minor comment to the authors

The effect of the manipulation of the FN-PF pathway is small and I am not really convinced this is a true effect on fear expression as compared to interindividual variability. I will tone down a bit this statement unless the authors show a fear expression effect on a subsequent retrieval test.

- We thank the reviewer for this observation and corrected the MS accordingly.

REVIEWERS' COMMENTS:

Reviewer #2 (Remarks to the Author):

In this manuscript the authors present a pathway specific contribution of a cerebellar (FN) to midbrain (vIPAG) to fear learning. They show that this projection represents a primarily excitatory channel of communication with neurons in both areas that are responsive to the CS and US during fear learning. Compellingly, they show that activation and inhibition of the activity of the FN-vIPAG pathway is able to bidirectionally modify the expression of freezing behavior to respectively attenuate and amplify the fear behavior during a multi-day extinction paradigm. Recordings of these areas during this excitatory perturbation, alongside new analyses that reveal that correlations with motor activity, reveal that activation of this connection is strengthened when this channel of communication is activated.

This paper is technically sound and is strengthened by the new results and anatomical quantification. In addition, the logic of the paper is improved, particularly in the first half of the manuscript. Overall the authors have addressed my primary concerns.

However, one recommendation is that the authors provide a clearer description of their proposed mechanism in the discussion. They do a good job of ruling out potential alternative hypotheses (including nociception, consolidation, and motor control), but the reader is left without a clear circuit-based mechanistic description of how activation of the FN results in this attenuated fear learning. A schematic that outlines how the circuit is operating during fear learning in non-perturbed conditions would help the reader tie together some of the disparate results.

REVIEWERS' COMMENTS:

Reviewer #2 (Remarks to the Author):

In this manuscript the authors present a pathway specific contribution of a cerebellar (FN) to midbrain (vIPAG) to fear learning. They show that this projection represents a primarily excitatory channel of communication with neurons in both areas that are responsive to the CS and US during fear learning. Compellingly, they show that activation and inhibition of the activity of the FN-vIPAG pathway is able to bidirectionally modify the expression of freezing behavior to respectively attenuate and amplify the fear behavior during a multi-day extinction paradigm. Recordings of these areas during this excitatory perturbation, alongside new analyses that reveal that correlations with motor activity, reveal that activation of this connection is strengthened when this channel of communication is activated.

This paper is technically sound and is strengthened by the new results and anatomical quantification. In addition, the logic of the paper is improved, particularly in the first half of the manuscript. Overall the authors have addressed my primary concerns.

However, one recommendation is that the authors provide a clearer description of their proposed mechanism in the discussion. They do a good job of ruling out potential alternative hypotheses (including nociception, consolidation, and motor control), but the reader is left without a clear circuit-based mechanistic description of how activation of the FN results in this attenuated fear learning. A schematic that outlines how the circuit is operating during fear learning in non-perturbed conditions would help the reader tie together some of the disparate results.

- Following the Reviewer suggestion, we have added a schematic representation of the circuit-based mechanism that explain that role of the FN in fear learning found in our study (Figure 9), based on our results and on the recent findings from other studies.